# PRIOR-GUIDED BAYESIAN OPTIMIZATION

## ABSTRACT

While Bayesian Optimization (BO) is a very popular method for optimizing expensive black-box functions, it fails to leverage the experience of domain experts. This causes BO to waste function evaluations on bad design choices (e.g., machine learning hyperparameters) that the expert already knows to work poorly. To address this issue, we introduce Prior-guided Bayesian Optimization (PrBO). PrBO allows users to inject their knowledge into the optimization process in the form of priors about which parts of the input space will yield the best performance, rather than BO's standard priors over functions (which are much less intuitive for users). PrBO then combines these priors with BO's standard probabilistic model to form a pseudo-posterior used to select which points to evaluate next. We show that PrBO is around 12x faster than state-of-the-art methods without user priors and $10,000\times$ faster than random search on a common suite of benchmarks, and achieves a new state-of-the-art performance on a real-world hardware design application. We also show that PrBO converges faster even if the user priors are not entirely accurate and that it robustly recovers from misleading priors.

## 1 INTRODUCTION

Bayesian Optimization (BO) is a data-efficient method for the joint optimization of design choices that gained great popularity in recent years. It is impacting a wide range of areas, including hyperparameter optimization (Snoek et al., 2012; Falkner et al., 2018), AutoML (Feurer et al., 2015a; Hutter et al., 2018), robotics (Calandra et al., 2016), computer vision (Nardi et al., 2017; Bodin et al., 2016), environmental monitoring (Marchant & Ramos, 2012), combinatorial optimization (Hutter et al., 2011), experimental design (Azimi et al., 2012), RL (Brochu et al., 2010), Computer Go (Chen et al., 2018), hardware design (Koeplinger et al., 2018; Nardi et al., 2019) and many others. It promises greater automation so as to increase both product quality and human productivity. As a result, BO is also established in many large tech companies, e.g., with Google Vizier (Golovin et al., 2017) and Facebook BoTorch (Balandat et al., 2019).

Nevertheless domain experts often have substantial prior knowledge that standard BO cannot incorporate. Users can incorporate prior knowledge by narrowing the search space; however, this type of hard prior can lead to poor performance by missing important regions. BO also supports a prior over functions $p(f)$, e.g., via a kernel function. However, this is not the prior experts have: users often know which ranges of hyperparameters tend to work best, and are able to specify a probability distribution $p_{\text{best}}(\boldsymbol{x})$ to quantify these priors. E.g., many users of the Adam optimizer (Kingma & Ba, 2015) know that its best learning rate is often in the vicinity of 1e-3 (give or take one order of magnitude), yet one may not know the accuracy one may achieve in a new application. Similarly, Navruzyan et al. (2019) derived neural network hyperparameter priors for image datasets based on their experience with five datasets. In these cases, users know potentially good values for a new application, but cannot be certain about them.

As a result, many competent users instead revert to manual search, which can fully incorporate their prior knowledge. A recent survey showed that most NeurIPS 2019 and ICLR 2020 papers that reported having tuned hyperparameters used manual search, with only a very small fraction using BO (Bouthillier & Varoquaux, 2020). In order for BO to be adopted widely, and help facilitate faster progress in the ML community by tuning hyperparameters faster and better, it is therefore crucial to devise a method that fully incorporates expert knowledge into BO. In this paper, we introduce Prior-guided Bayesian Optimization (PrBO), a novel BO variant that combines user prior knowledge with a probabilistic model of the observations made. Our technical contributions with PrBO are:

1. PrBO bridges the TPE methodology and standard BO probabilistic models, such as GPs, RFs or Bayesian NNs, instead of Tree Parzen Estimators only.

2. PrBO is flexible w.r.t. how the prior is defined, allowing previously hard-to-inject (e.g. exponential) priors.

3. PrBO gives more importance to the model as iterations progress, gradually forgetting the prior and ensuring robustness against misleading priors.

We demonstrate the effectiveness of PrBO on a comprehensive set of real-world applications and synthetic benchmarks, showing that accurate prior knowledge helps PrBO to achieve similar performance to current state-of-the-art on average $12.12\times$ faster on synthetic benchmarks and $1.49\times$ faster on a real-world application. PrBO also achieves equal or better final performance in all but one of the benchmarks tested.

## 2    BACKGROUND

### 2.1    BAYESIAN OPTIMIZATION

Bayesian Optimization (BO) is an approach for optimizing an unknown function $f : \mathcal{X} \to \mathbb{R}$ that is expensive to evaluate over an input space $\mathcal{X}$. In this paper, we aim to minimize $f$, i.e., find $\boldsymbol{x}^* \in \arg\min_{\boldsymbol{x}\in\mathcal{X}} f(\boldsymbol{x})$. BO approximates $\boldsymbol{x}^*$ with an optimal sequence of evaluations $\boldsymbol{x}_1, \boldsymbol{x}_2, \ldots \in \mathcal{X}$, with each new $\boldsymbol{x}_{n+1}$ depending on the previous function values $y_1, y_2, \ldots, y_n$ at $\boldsymbol{x}_1, \ldots, \boldsymbol{x}_n$. BO achieves this by building a posterior on $f$ based on the set of evaluated points. At each BO iteration, a new point is selected and evaluated based on the posterior, and the posterior is updated to include the new point $(\boldsymbol{x}_{n+1}, y_{n+1})$.

The points explored by BO are dictated by the acquisition function, which attributes a value to each $\boldsymbol{x} \in \mathcal{X}$ by balancing the predicted value and uncertainty of the prediction for each $\boldsymbol{x}$. In this work, as the acquisition function we choose Expected Improvement (EI) (Mockus et al., 1978), which quantifies the expected improvement over the best function value found so far: $EI_{f_{inc}}(\boldsymbol{x}) := \int_{-\inf}^{\inf} \max(f_{inc} - y, 0)p(y|\boldsymbol{x})dy$, where $f_{inc}$ is the incumbent function value, i.e., the best objective function value found so far, and $p(y|\boldsymbol{x})$ is a probabilistic model, e.g., a GP. Alternatives to EI would be Probability of Improvement (PI) (Jones, 2001), upper-confidence bounds (UCB) (Srinivas et al., 2010), and entropy-based methods (e.g. Hernández-Lobato et al. (2014)).

### 2.2    TREE-STRUCTURED PARZEN ESTIMATOR

Whereas the standard probabilistic model in BO directly models $p(y|\boldsymbol{x})$, the Tree-structured Parzen Estimator (TPE) approach of Bergstra et al. (2011) models $p(\boldsymbol{x}|y)$ and $p(y)$ instead[1]. This is done by constructing two parametric densities, $g(\boldsymbol{x})$ and $l(\boldsymbol{x})$, which are computed using the observations with function value above and below a given threshold, respectively. The separating threshold $y^*$ is defined as a quantile of the observed function values. TPE uses the densities $g(\boldsymbol{x})$ and $l(\boldsymbol{x})$ to define $p(\boldsymbol{x}|y)$ as:

$$p(\boldsymbol{x}|y) = l(\boldsymbol{x})I(y < y^*) + g(\boldsymbol{x})(1 - I(y < y^*)), \tag{1}$$

where $I(y < y^*)$ is 1 when $y < y^*$ and 0 otherwise. The parametrization of the generative model $p(\boldsymbol{x}, y) = p(\boldsymbol{x}|y)p(y)$ facilitates the computation of EI as it leads to $EI_{y^*}(\boldsymbol{x}) \propto l(\boldsymbol{x})/g(\boldsymbol{x})$ and, thus, $\arg\max_{\boldsymbol{x}\in\mathcal{X}} EI_{y^*}(\boldsymbol{x}) = \arg\max_{\boldsymbol{x}\in\mathcal{X}} l(\boldsymbol{x})/g(\boldsymbol{x})$.

## 3    BAYESIAN OPTIMIZATION WITH PRIORS

We propose a BO approach dubbed PrBO that allows field experts to inject user prior knowledge into the optimization in the form of priors. PrBO combines this user-defined prior with a probabilistic

---

[1]Note that, technically, the model does not parameterize $p(y)$, since it is computed based on the observed data points, which are heavily biased towards low values due to the optimization process. Instead, it parameterizes a dynamically changing $p(y_i)_{i=1}^t$, which helps to constantly challenge the model to yield better observations.

model that captures the likelihood of the observed data $(\boldsymbol{x}_i, y_i)_{i=1}^t$. PrBO is independent from the probabilistic model being used, i.e., it can be freely combined with, e.g., GPs, RFs, or Bayesian Neural Networks.

## 3.1 PRIORS

PrBO allows users to inject prior knowledge into BO. This is done via a prior distribution that informs where in the input space $\mathcal{X}$ we expect to find good $f(\boldsymbol{x})$ values. A point is considered "good" if it leads to low function values. We denote the prior distribution $P_g(\boldsymbol{x})$, where $g$ denotes that this is a prior on good points and $\boldsymbol{x} \in \mathcal{X}$ is a given point. Similarly, we define a prior on where in the input space we expect to have "bad" points. Although we could have a separate user-defined probability distribution $P_b(\boldsymbol{x})$, we aimed to keep the load on users low and thus, for simplicity, only require the definition of $P_g(\boldsymbol{x})$ and compute $P_b(\boldsymbol{x}) = 1 - P_g(\boldsymbol{x})$.[2] $P_g(\boldsymbol{x})$ is normalized to $[0, 1]$ by min-max scaling before computing $P_b(\boldsymbol{x})$.

In practice, $\boldsymbol{x}$ contains several dimensions but it is difficult for domain experts to provide a multivariate distribution $P_g(\boldsymbol{x})$. Users can easily specify, e.g., draw, a univariate or bivariate probability distribution for continuous dimensions or provide a list of probabilities for discrete dimensions. In PrBO, users are free to define a complex multivariate distribution, but we expect the standard use case to be that users only want to specify univariate distributions, implicitly assuming a prior that factors as $P_g(\boldsymbol{x}) = \prod_{i=1}^D P_g(x_i)$, where $D$ is the number of dimensions in $\mathcal{X}$ and $x_i$ is the i-th input dimension of $\mathcal{X}$. We show examples of continuous and discrete priors in Appendices B and F, respectively. We use factorized priors in our experiments to mimic what we expect most users will provide. In Appendix G, we show that these factorized priors lead to similar performance compared to multivariate priors.

## 3.2 MODEL

Whereas the standard probabilistic model in BO, e.g., a GP, quantifies $p(y|\boldsymbol{x})$ directly, that model is hard to combine with the user-defined prior $P_g(\boldsymbol{x})$. We therefore now introduce a method to translate the standard probabilistic model $p(y|\boldsymbol{x})$ into a model that is easier to combine with this prior. Similar to the TPE work described in Sec. 2.2, our generative model combines $p(\boldsymbol{x}|y)$ and $p(y)$ instead of directly modeling $p(y|\boldsymbol{x})$.

The computation we perform for this translation is to quantify the probability that a given input $\boldsymbol{x}$ is "good" under our standard probabilistic model $p(y|\boldsymbol{x})$. As in TPE, we define configurations as "good" if their observed $y$-value is below a certain quantile $\gamma$ of the observed function values (so that $p(y < f_\gamma) = \gamma$). We in addition exploit the fact that our standard probabilistic model $p(y|\boldsymbol{x})$ has a Gaussian form, and under this Gaussian prediction we can compute the probability $\mathcal{M}_g(\boldsymbol{x})$ of the function value lying below a certain quantile using the standard closed-form formula for PI (Kushner, 1964):

$$\mathcal{M}_g(\boldsymbol{x}) = p(f(\boldsymbol{x}) < f_\gamma | \boldsymbol{x}, (\boldsymbol{x}_i, y_i)_{i=1}^t) = \Phi\left(\frac{f_\gamma - \mu_{\boldsymbol{x}}}{\sigma_{\boldsymbol{x}}}\right), \qquad (2)$$

where $(\boldsymbol{x}_i, y_i)_{i=1}^t$ are the evaluated configurations and their function value, $\mu_{\boldsymbol{x}}$ and $\sigma_{\boldsymbol{x}}$ are the mean and standard deviation of the probabilistic model at $\boldsymbol{x}$, and $\Phi$ is the standard normal CDF, see Figure 1. Note that there are two probabilistic models here:

- The standard probabilistic model of BO, with a prior over functions $p(f)$, updated by data $(\boldsymbol{x}_i, y_i)_{i=1}^t$ to yield a posterior over functions $p(f|(\boldsymbol{x}_i, y_i)_{i=1}^t)$, allowing us to quantify the probability $\mathcal{M}_g(\boldsymbol{x}) = p(f(x) < f_\gamma | \boldsymbol{x}, (\boldsymbol{x}_i, y_i)_{i=1}^t)$ in Equation 2
- The TPE-like generative model that combines $p(y)$ and $p(\boldsymbol{x}|y)$ instead of directly modelling $p(y|\boldsymbol{x})$.

Equation 2 bridges these two models by using the probability of improvement from BO's standard model as the probability $\mathcal{M}_g(\boldsymbol{x})$ in TPE's model. Ultimately, this is a heuristic since there is no formal

---

[2]We note that for continuous spaces, this $P_b(\boldsymbol{x})$ is not a probability distribution, and therefore only a pseudo-prior, as it does not integrate to 1. For discrete spaces, we normalize $P_b(\boldsymbol{x})$ so that it sums to 1 and therefore is a proper probability distribution and prior.

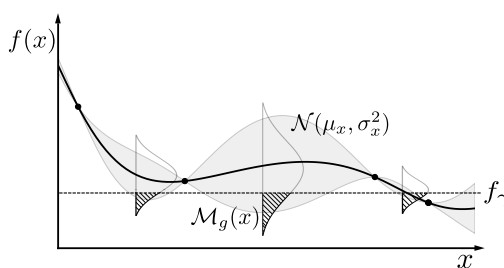

Figure 1: Our model is composed by a probabilistic model and the probability of improving over the threshold $f_\gamma$, i.e., right tail of the Gaussian. The black curve is the probabilistic model's mean and the shaded area is the model's variance.

**Algorithm 1** PrBO Algorithm. $\boldsymbol{D}$ keeps track of all function evaluations so far: $(\boldsymbol{x}_i, y_i)_{i=1}^t$.

1: **Input:** Input space $\mathcal{X}$, user-defined prior distributions $P_g(\boldsymbol{x})$ and $P_b(\boldsymbol{x})$, quantile $\gamma$ and BO budget $B$.
2: **Output:** Optimized point $\boldsymbol{x}_{inc}$.
3: $\boldsymbol{D} \leftarrow Initialize(\mathcal{X})$
4: **for** $t = 1$ **to** $B$ **do**
5: $\quad \mathcal{M}_g(\boldsymbol{x}) \leftarrow fit\_model\_good(\boldsymbol{D})$
6: $\quad \mathcal{M}_b(\boldsymbol{x}) \leftarrow fit\_model\_bad(\boldsymbol{D})$
7: $\quad g(\boldsymbol{x}) \leftarrow P_g(\boldsymbol{x}) \cdot \mathcal{M}_g(\boldsymbol{x})^{\frac{t}{\beta}}$
8: $\quad b(\boldsymbol{x}) \leftarrow P_b(\boldsymbol{x}) \cdot \mathcal{M}_b(\boldsymbol{x})^{\frac{t}{\beta}}$
9: $\quad \boldsymbol{x}_t \in \arg\max_{\boldsymbol{x} \in \mathcal{X}} EI_{f_\gamma}(\boldsymbol{x})$
10: $\quad y_t \leftarrow f(\boldsymbol{x}_t)$
11: $\quad \boldsymbol{D} = \boldsymbol{D} \cup (\boldsymbol{x}_t, y_t)$
12: **end for**
13: $\boldsymbol{x}_{inc} \leftarrow ComputeBest(\boldsymbol{D})$
14: **return** $\boldsymbol{x}_{inc}$

connection between the two probabilistic models. However, we believe that the use of BO's familiar, theoretically sound framework of probabilistic modelling of $p(y|x)$, followed by the computation of the familiar PI formula is a very intuitive choice for obtaining the probability of an input achieving at least a given performance threshold – exactly the term we need for TPE's $\mathcal{M}_g(\boldsymbol{x})$. Similarly, we also define a probability $\mathcal{M}_b(\boldsymbol{x})$ of $\boldsymbol{x}$ being bad as $\mathcal{M}_b(\boldsymbol{x}) = 1 - \mathcal{M}_g(\boldsymbol{x})$.

### 3.3 PSEUDO-POSTERIOR

PrBO combines the prior in Section (3.1) and the model in Eq. (2) into a pseudo-posterior on "good" points. This pseudo-posterior represents the updated beliefs on where we can find good points, based on the prior and data that has been observed. The pseudo-posterior is computed as the product of the prior and the model:

$$g(\boldsymbol{x}) \propto P_g(\boldsymbol{x}) \mathcal{M}_g(\boldsymbol{x})^{\frac{t}{\beta}}, \tag{3}$$

where $t$ is the current optimization iteration, $\beta$ is an optimization hyperparameter, $\mathcal{M}_g(\boldsymbol{x})$ is defined in Eq. (2), and $P_g(\boldsymbol{x})$ is the prior defined in Sec 3.1, rescaled to $[0, 1]$. We note that this pseudo-posterior is not normalized, but this suffices for PrBO to determine the next $\boldsymbol{x}_t$ as the normalization constant cancels out (c.f. Section 3.4). Since $g(\boldsymbol{x})$ is not normalized and we include the exponent $t/\beta$ in Eq. 3, we refer to $g(\boldsymbol{x})$ as a pseudo-posterior, to emphasize that it is not a standard posterior probability distribution.

The $t/\beta$ fraction in Eq. (3) controls how much weight is given to the model. As the optimization progresses, more weight is given to the model over the prior. Intuitively, we put more emphasis on the model as it observes more data and becomes more accurate. We do this under the assumption that the model will eventually be better than the user at predicting where to find good points. This also allows to recover from misleading priors as we show in Section 4.2; similar to, and inspired by Bayesian models, the data ultimately washes out the prior. The $\beta$ hyperparameter defines the balance between prior and model, with higher $\beta$ values giving more importance to the prior and requiring more data to overrule it.

We note that, computing Equation (3) directly can lead to numerical issues. Namely, the pseudo-posterior can reach extremely low values if the prior and model probabilities are low, especially as the $t/\beta$ exponent grows. To prevent this, in practice, PrBO uses the logarithm of the pseudo-posterior instead: $\log(g(\boldsymbol{x})) \propto \log(P_g(\boldsymbol{x})) + \frac{t}{\beta} \cdot \log(\mathcal{M}_g(\boldsymbol{x}))$.

Once again, we also define an analogous pseudo-posterior distribution on bad $\boldsymbol{x}$: $b(\boldsymbol{x}) \propto P_b(\boldsymbol{x})\mathcal{M}_b(\boldsymbol{x})^{\frac{t}{\beta}}$. We then use these quantities to define a density model $p(\boldsymbol{x}|y)$ as follows:

$$p(\boldsymbol{x}|y) \propto \begin{cases} g(\boldsymbol{x}) & \text{if } y < f_\gamma \\ b(\boldsymbol{x}) & \text{if } y \geq f_\gamma. \end{cases} \tag{4}$$

## 3.4 Acquisition Function

We adopt the EI formulation used in (Bergstra et al., 2011) by replacing their Adaptive Parzen Estimators with our computation of the pseudo-posterior in Eq. (3). Namely, we compute EI as:

$$EI_{f_\gamma}(\boldsymbol{x}) := \int_{-\inf}^{\inf} \max(f_\gamma - y, 0)p(y|\boldsymbol{x})dy = \int_{-\inf}^{f_\gamma} (f_\gamma - y)\frac{p(\boldsymbol{x}|y)p(y)}{p(\boldsymbol{x})}dy \propto \left(\gamma + \frac{b(\boldsymbol{x})}{g(\boldsymbol{x})}(1 - \gamma)\right)^{-1}. \tag{5}$$

The full derivation of Eq. (5) is shown in Appendix C. Eq. (5) shows that to maximize improvement we would like points $\boldsymbol{x}$ with high probability under $g(\boldsymbol{x})$ and low probability under $b(\boldsymbol{x})$, i.e., minimizing the ratio $b(\boldsymbol{x})/g(\boldsymbol{x})$. We note that the point that minimizes the ratio for our unnormalized pseudo-posteriors will be the same that minimizes the ratio for the normalized pseudo-posterior and, thus, the computation of the normalized pseudo-posteriors is unnecessary.

The dynamics of the PrBO algorithm can be understood in terms of the following proposition (proof in Appendix C):

**Proposition 1** *Given $f_\gamma$, $P_g(\boldsymbol{x})$, $P_b(\boldsymbol{x})$, $\mathcal{M}_g(\boldsymbol{x})$, $\mathcal{M}_b(\boldsymbol{x})$, $g(\boldsymbol{x})$, $b(\boldsymbol{x})$, $p(\boldsymbol{x}|y)$, and $\beta$ as above, then*

$$\lim_{t\to\infty} \arg\max_{\boldsymbol{x}\in\mathcal{X}} EI_{f_\gamma}(\boldsymbol{x}) = \lim_{t\to\infty} \arg\max_{\boldsymbol{x}\in\mathcal{X}} \mathcal{M}_g(\boldsymbol{x}),$$

*where $EI_{f_\gamma}$ is the Expected Improvement acquisition function as defined in Eq. 5 and $\mathcal{M}_g(\boldsymbol{x})$ is as defined in Equation (2).*

In early BO iterations the prior will have a predominant role, but in later BO iterations the model will grow more important, and as Proposition 1 shows, if PrBO is run long enough the prior washes out and PrBO *only* trusts the probabilistic model informed by the data.

## 3.5 Putting It All Together

Algorithm 1 shows the PrBO algorithm, based on the components defined in the previous sections. In Line 3, PrBO starts with a design of experiments (DoE) phase, where it randomly samples a number of points from the user-defined prior $P_g(\boldsymbol{x})$. After initialization, the BO loop starts at Line 4. In each loop iteration, PrBO fits the probabilistic model on the previously evaluated points (lines 5 and 6) and computes the pseudo-posteriors $g(\boldsymbol{x})$ and $b(\boldsymbol{x})$ (lines 7 and 8 respectively). The EI acquisition function is computed next, using the pseudo-posteriors, and the point that maximizes EI is selected as the next point to evaluate at line 9. The black-box function evaluation is performed at Line 10. This BO loop is repeated for a pre-defined number of iterations, according to the user-defined budget $B$. In Appendix A, we show a visualization of PrBO on a 1-dimensional function.

## 4 Experiments

We implement both GPs and RFs as predictive models and use GPs in all experiments, except for our real-world experiments (Section 4.4) where we use RFs for a fair comparison. We set the model weight $\beta = 10$ and the model quantile to $\gamma = 0.05$, see our sensitivity hyperparameter study in Appendices L and M. Before starting the main BO loop in PrBO, we randomly sample $D + 1$ points from the prior consistently on all benchmarks. In Appendix J, we compare PrBO to other baselines with the same prior initialization and show that it leads to similar results. We optimize our EI acquisition function using a multi-start local search, similar to SMAC (Hutter et al., 2011). We start with four synthetic benchmarks: Branin, SVM, FC-Net, and XGBoost, which are 2, 2, 6, and 8 dimensional, respectively. The last three are part of the Profet benchmarks (Klein et al., 2019), generated by a generative model built using performance data on OpenML or UCI datasets. See

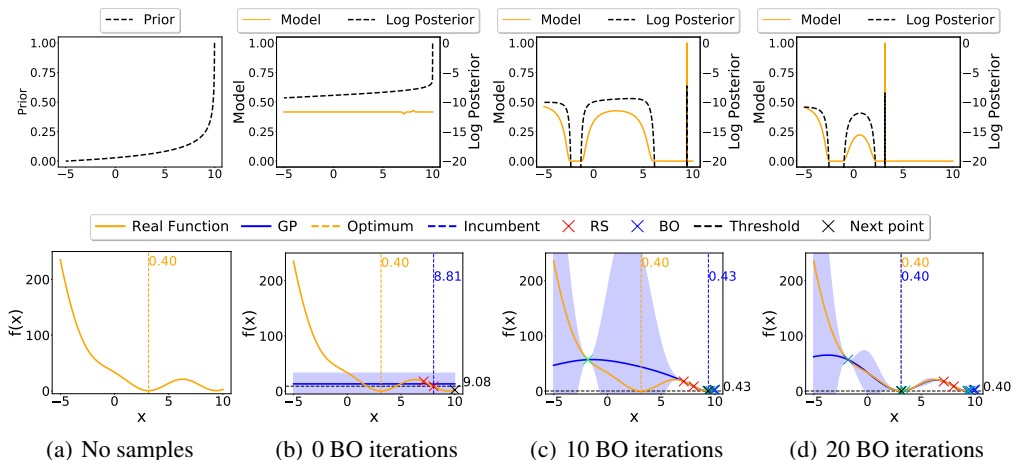

Figure 2: PrBO on the 1D Branin function. The leftmost column shows the exponential prior. The other columns show the model and the log pseudo-posterior after 0 (RS only), 10, and 20 BO iterations. PrBO forgets the wrong prior on the local optimum and converges to the global optimum.

Appendix D for more details on the experimental setup. Due to space constraints, we defer the (qualitatively similar) results for the SVM benchmark to Appendix E.

### 4.1 PRIOR SELECTION

In this section we study the effect of choosing a prior. A suitable property of the prior is that, by selecting a tighter prior around an optimum, we would expect sampling from the prior to have an increased performance. To the limit, if the prior is composed by only one point which is one of the global optima, then the first sample (and all of them) from the prior will hit the optimum. To have a sanity check of this property, we build an artificial prior in a controlled way. We rely on an automated computation of the prior by computing a univariate Kernel Density Estimation (KDE) using a Gaussian kernel on the synthetic benchmarks introduced above. We note that the goal of these synthetic priors is to have an unbiased prior for our experiments, whereas manual priors would be biased by our own expertise of these benchmarks. In practice, users will manually define these priors without needing additional experiments. Appendix G shows that using univariate KDEs for the prior lead to similar performance compared to multivariate KDEs.

We experiment with an array of varying quality priors. We select a constant $10D$ points in each prior and vary the size of the random sample dataset so that we can make the priors more sharply peaked around the optima in a controlled environment. We use the best performing $10D$ samples to create the prior from a uniform random sample dataset size of $10D\frac{100}{x}$; we refer to this prior as $x\%$ in Figure 3. As an example the XGBoost benchmark has $d = 8$, so, $100\%$ means we sample 80 points and use all 80 to create the prior, $10\%$ means we sample 800 points and use the best performing 80 to create the prior, $1\%$ means we sample $8,000$ and use the best 80 to create the prior, and so on.

Figure 3 shows the performance of purely sampling from the prior and running PrBO, respectively, after $10D$ function evaluations with different priors. A bigger random sample dataset and a smaller percentage leads to a tighter prior around the optimum, making the argument for a stronger prior. This is confirmed by Figure 3, where a sharply peaked prior (right side of the figure) leads to a better performance in both scenarios. In addition we observe that in contrast to sampling from the prior, PrBO achieves a smaller regret by being able to evolve from the initial prior and making independent steps towards better values of the objective function. More extensive experiments with a similar trend, including the rest of the benchmarks, are in Appendix K.

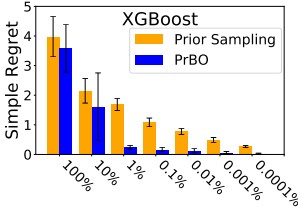

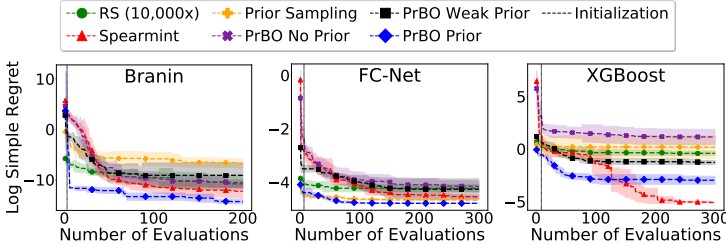

Figure 3: Regret of prior sampling and PrBO with different priors ($\mu \pm \sigma$ on 5 reps.).

Figure 4: Log regret comparison of PrBO with and without priors, RS, and Spearmint (mean +/- one std on 5 repetitions). We run the benchmarks for $100D$ iterations, capped at 300.

## 4.2 PRIOR FORGETTING

In this section, we show that PrBO can recover from a misleading prior, thanks to our predictive model and the $t/\beta$ parameter in the pseudo-posterior computation Eq. 3. As BO progresses, the predictive model becomes more accurate and receives more weight, guiding optimization away from the wrong prior and towards better values of the objective function. Figure 2 shows PrBO on the 1D Branin function with an exponential prior. Columns (b), (c), and (d) show PrBO after $D + 1 = 2$ initial samples and 0, 10, 20 BO iterations, respectively. After initialization, as shown in Column (b), the pseudo-posterior is nearly identical to the exponential prior and guides PrBO towards the region of the space on the right, which is towards the local optimum. This happens until the predictive model becomes certain there will be no more improvement from sampling that region (Columns (c) and (d)). After that, the predictive model guides the pseudo-posterior towards exploring regions with high uncertainty. Once the global minimum region is found, the pseudo-posterior starts balancing exploiting the global minimum and exploring regions with high uncertainty, as shown in 2d (bottom). Notably, the pseudo-posterior after $x > 4$ falls to 0 in 2d (top), as the predictive model is certain there will be no improvement from sampling the region of the local optimum. We provide additional examples of forgetting in Appendix B, and a comparison of PrBO with misleading priors, no prior, and correct priors in Appendix H.

## 4.3 COMPARISON AGAINST STRONG BASELINES

Figure 4 compares PrBO to other optimizers using the log simple regret on five runs (mean and std error reported) on the synthetic benchmarks. We consider two priors in our experiments, a strong prior, computed using a KDE on the best $10D$ out of $10,000,000D$ uniform random samples, and a weak prior, computed using a KDE on the best $10D$ out of $1,000D$ uniform random samples. We emphasize that we only used these artificial priors in these experiments to guarantee that our prior is not biased by our own expertise for the benchmarks we used. In practice, the prior is defined by the user. We compare the results of PrBO with and without priors (both weak and strong) to $10,000\times$ random search (RS, i.e., for each BO sample we draw $10,000$ uniform random samples), sampling from the strong prior only, and Spearmint (Snoek et al., 2012) which is a well-adopted BO approach using GPs and the EI acquisition function. In Appendix I we show a comparison of PrBO to TPE and SMAC.

PrBO Prior beats $10,000\times$ RS and PrBO weak prior on all benchmarks. It also outperforms the performance of sampling from the prior; this is expected because prior sampling cannot recover from a non-ideal prior. The two methods are identical up to the initialization phase because they both sample from the same prior in that phase.

PrBO Prior is more sample efficient and finds better or equal results than Spearmint on three out of the four benchmarks. On the fourth benchmark, XGBoost, PrBO leads the performance until 139 BO iterations, where Spearmint catches up and achieves slightly better results in the end (function values of $8.986$ for Spearmint vs. $9.026$ for PrBO after 300 iterations). Importantly, in all our experiments, PrBO with a good prior consistently shows tremendous speedups in the early phases of the optimization process, typically only requiring on average $8.25$ iterations to reach the performance that Spearmint reaches after 100 iterations ($12.12\times$ faster). Thus, in comparison to other traditional

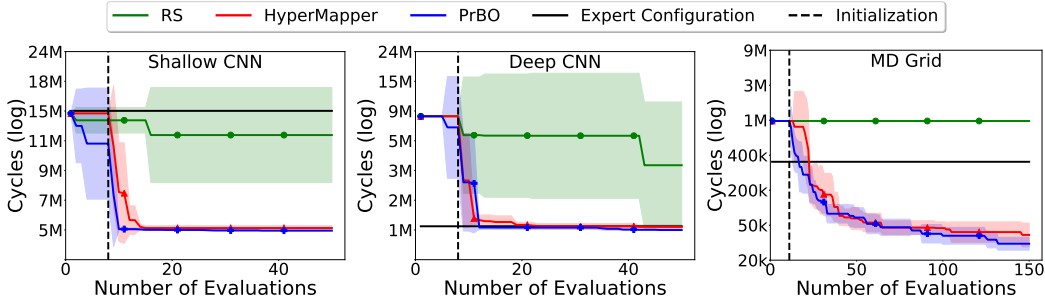

Figure 5: Log regret comparison of RS, HyperMapper, PrBO, and manual optimization on Spatial. The line and shaded regions show mean and std after 5 repetitions. Vertical lines are initialization.

BO approaches, PrBO makes use of the best of both worlds, leveraging prior knowledge and efficient optimization based on BO.

### 4.4 THE SPATIAL USE-CASE

We next apply PrBO to the `Spatial` (Koeplinger et al., 2018) real-world application. `Spatial` is a programming language and corresponding compiler for the design of application accelerators, i.e., FPGAs. We apply PrBO to three `Spatial` benchmarks, namely, 7D shallow and deep CNNs, and a 10D molecular dynamics grid application. We compare the performance of PrBO to RS, manual optimization, and HyperMapper (Nardi et al., 2019), the current state-of-the-art BO solution for `Spatial`. For a fair comparison, both PrBO and HyperMapper use RFs as predictive probabilistic model. The manual optimization and the prior for PrBO were provided by an unbiased `Spatial` developer, who is not an author of this paper. The priors were provided once and kept unchanged for the whole project. More details on the setup, including the priors used, are presented in Appendix F.

Figure 5 shows the log regret on the `Spatial` benchmarks. PrBO vastly outperforms RS in all benchmarks; notably, RS does not improve over the default configuration in MD Grid. PrBO is also able to leverage the expert's prior and outperform the expert's configuration in all benchmarks ($2.68\times$, $1.06\times$, and $10.4\times$ speedup for shallow CNN, deep CNN, and MD Grid, respectively). Compared to HyperMapper, PrBO achieves better performance in the MD Grid benchmark ($1.28\times$ speedup). For context, this is a significant improvement in the FPGA field, where a 10% improvement could qualify for acceptance in a top-tier conference. In the CNN benchmarks, PrBO converges to the minima regions faster than HyperMapper ($1.58\times$ and $1.4\times$ faster for shallow and deep respectively). Thus, PrBO leverages the best of both worlds (the expert's prior knowledge and BO) to provide a new state of the art for `Spatial`.

## 5 RELATED WORK

TPE by Bergstra et al. (2011), the default optimizer in the popular HyperOpt package (Bergstra et al., 2013), supports limited hand-designed priors in the form of normal or log-normal distributions. We make three technical contributions that make PrBO more flexible than TPE. First, we generalize over the TPE approach by allowing more flexible priors; second, our approach is model-agnostic (i.e., PrBO is not limited to the TPE model; we use GPs and RFs in our experiments); and third, PrBO is inspired by Bayesian models that give more importance to the data as iterations progress. We also show that PrBO outperforms HyperOpt's TPE in Appendix I.

In parallel work, Li et al. (2020) allow users to specify priors via a probability distribution, similarly to our approach. Their two-level approach first samples a number of configurations by maximizing Thompson samples from a GP pseudo-posterior and then chooses the configuration that has the highest prior as the next to evaluate. In contrast, our method leverages the information from the prior more directly; is agnostic to the probabilistic model used, which is important for applications with many discrete variables like our real-world application, where RFs outperform GPs; and provably

recovers gracefully from misspecified priors, while the approach of Li et al. (2020) fundamentally trusts the prior and it never gets washed out by more data..

The work of Ramachandran et al. (2020) also supports priors in the form of probability distributions. Their work uses the probability integral transform to warp the search space, stretching regions where the prior has high probability, and shrinking others. Once again, compared to their approach, PrBO is agnostic to the probabilistic model used and directly controls the balance between prior and model via the $\beta$ hyperparameter. Additionally, PrBO's probabilistic model is fitted independently from the prior, which ensures it is not biased by the prior, while their approach fits the model to a warped version of the space, transformed using the prior, making it more difficult to recover from misleading priors.

Black-box optimization tools, such as SMAC (Hutter et al., 2011) or iRace (López-Ibáñez et al., 2016) also support simple hand-designed priors, e.g. log-transformations. However, these are not properly reflected in the predictive models and both cannot explicitly recover from bad priors.

Oh et al. (2018) and Siivola et al. (2018) propose structural priors for high-dimensional problems. They assume that users always place regions they expect to be good at the center of the search space and then develop BO approaches that favor configurations near the center. However, this is a rigid assumption about optimum locality, which does not allow users to freely specify their priors.

Similarly, Shahriari et al. (2016) focus on unbounded search spaces. The priors in their work are not about good regions of the space, but rather a regularization function that penalizes configurations based on their distance to the center of the user-defined search space. The priors are automatically derived from the search space and users may not even be aware of the priors that are used.

Our work also relates to meta-learning for BO (Vanschoren, 2019), where BO is applied to many similar optimization problems in a sequence such that knowledge about the general problem structure can be exploited in future optimization problems. In contrast to meta-learning, PrBO is the first method that allows human experts to explicitly specify their priors. Furthermore, PrBO does not depend on any meta-features (Feurer et al., 2015b) and only incorporates a single prior instead of many priors from different experiments (Lindauer & Hutter, 2018).

## 6 Conclusions and Future Work

We have proposed a novel BO variant, PrBO, that allows users to inject their expert knowledge into the optimization in the form of priors about which parts of the input space will yield the best performance. These are different than standard priors over functions which are much less intuitive for users. BO failed so far to leverage the experience of domain experts, not only causing inefficiency but also getting users away from applying BO approaches because they could not exploit their years of knowledge in optimizing their black-box functions. PrBO addresses this issue and will therefore facilitate the adoption of BO. We showed that PrBO is 12.12x more sample efficient than state-of-the-art methods, and $10,000\times$ faster than random search, on a common suite of benchmarks and achieves a new state-of-the-art performance on a real-world hardware design application. We also showed that PrBO converges faster and that it robustly recovers from misleading priors.

In future work, we will study how our approach can be used to leverage prior knowledge from meta-learning. Bringing these two worlds together will likely boost the performance of BO even further.

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

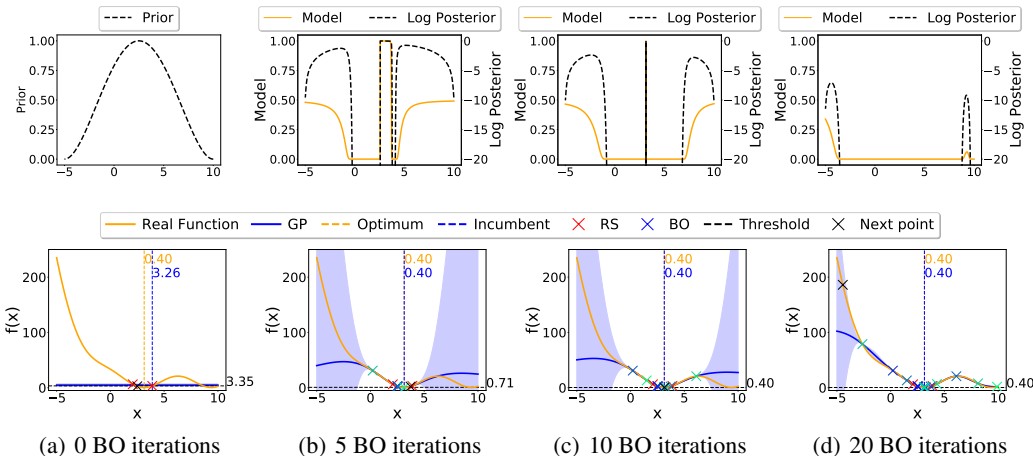

(a) 0 BO iterations     (b) 5 BO iterations     (c) 10 BO iterations     (d) 20 BO iterations

Figure 6: Breakdown of the prior $P_g(\boldsymbol{x}) = \mathcal{B}(3,3)$, the model $M_g(\boldsymbol{x})$, and the pseudo-posterior $g(\boldsymbol{x})$ (top row) for the 1-dimensional Branin function (bottom row) and their evolution over the optimization iterations. Two random points from the prior are sampled to initialize the GP model (bottom row) before starting PrBO. The blue/green crosses denote PrBO samples, with green samples denoting later iterations. The blue line and shaded area show the mean and uncertainty of the GP model.

## A  MODEL AND PSEUDO-POSTERIOR VISUALIZATION

This section visualizes the prior $P_g(\boldsymbol{x})$, the model $M_g(\boldsymbol{x})$, and the pseudo-posterior $g(\boldsymbol{x})$ for a 1-dimensional Branin function and their evolution over the optimization iterations. We define this function by setting the second dimension of the Branin function to the global optimum $x_2 = 2.275$ and optimizing the first dimension. We perform an initial design of $D+1 = 2$ random points sampled from the prior and use a GP as predictive model. We use a Beta distribution prior $P_g(\boldsymbol{x}) = \mathcal{B}(3,3)$ which resembles a truncated Gaussian centered close to the global optimum and compute the model $\mathcal{M}_g(\boldsymbol{x})$ and pseudo-posterior $g(\boldsymbol{x})$ following Eq. (2) and (3) respectively. Figure 6 shows the optimization at different stages.

Figure 6a shows the initialization phase (bottom) and the Beta prior (top). After 5 BO iterations, in Figure 6b (top), the pseudo-posterior is high near the global minimum, around $x = \pi$, where both the prior and the model agree there are good points. After 10 BO iterations in Figure 6c (top), there are three regions with high pseudo-posterior. The middle region, where PrBO is exploiting until the optimum is found, and two regions to the right and left, which will lead to future exploration as shown in Figure 6d (bottom) on the right and left of the global optimum in light green crosses. After 20 iterations, see Figure 6d (top), the pseudo-posterior vanishes where the model is certain there will be no improvement, but it is high wherever there is uncertainty in the GP. Note that the influence of the prior after 20 iterations is weaker, because of $t/\beta$ in Eq. (3).

## B  PRIOR FORGETTING SUPPLEMENTARY EXPERIMENTS

In this section, we show additional evidence that PrBO can recover from wrongly defined priors so to complement section 4.2. Figure 7 shows PrBO on the 1D Branin function as in Figure 2 but with a decay prior. Column (a) of Figure 7 shows the decay prior and the 1D Branin function. This prior emphasizes the wrong belief that the optimum is likely located on the left side around $x = -5$ while the optimum is located at the orange dashed line. Columns (b), (c), and (d) of Figure 7 show PrBO on the 1D Branin after $D + 1 = 2$ initial samples and $0$, $10$, and $20$ BO iterations, respectively. In the beginning of BO, as shown in column (b), the pseudo-posterior is nearly identical to the prior and guides PrBO towards the left region of the space. As more points are sampled, the model becomes more accurate and starts guiding the pseudo-posterior away from the wrong prior (column (c)). Notably, the pseudo-posterior before $x = 0$ falls to 0, as the predictive model is certain there will

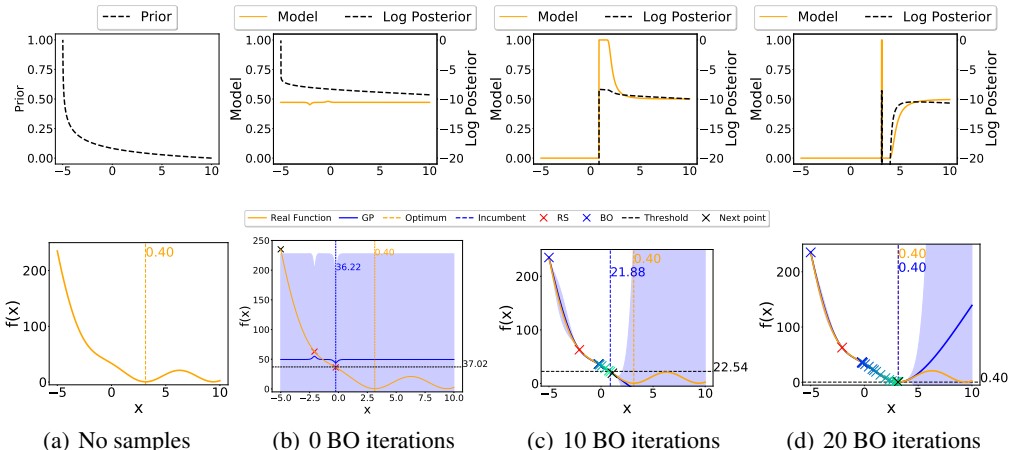

Figure 7: PrBO on the 1D Branin function with a decay prior. The leftmost column shows the log pseudo-posterior before any samples are evaluated, in this case, the pseudo-posterior is equal to the decay prior. The other columns show the model and pseudo-posterior after 0 (only random samples), 10, and 20 BO iterations. 2 random samples are used to initialize the GP model.

be no improvement from sampling this region. After 20 iterations, PrBO finds the optimum region, despite the poor start (column (d)). The peak in the pseudo-posterior in column (d) shows PrBO will continue to exploit the optimum region as it is not certain if the exact optimum has been found. The pseudo-posterior is also high in the high uncertainty region after $x = 4$, showing PrBO will explore that region after it finds the optimum.

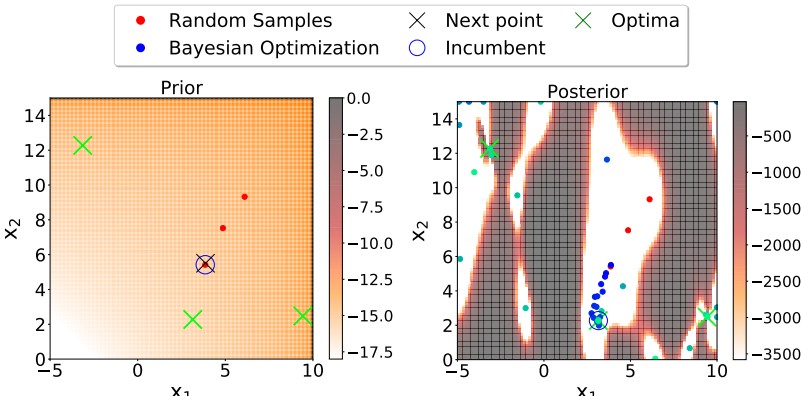

Figure 8: PrBO on the Branin function with exponential priors for both dimensions. (a) shows the log pseudo-posterior before any samples are evaluated, in this case, the pseudo-posterior is equal to the prior; the green crosses are the optima. (b) shows the result of optimization after 3 initialization samples drawn from the prior at random and 50 BO iterations. The dots in (b) show the points explored by PrBO, with greener points denoting later iterations. The colored heatmap shows the log of the pseudo-posterior $g(\boldsymbol{x})$.

Figure 8 shows PrBO on the standard 2D Branin function. We use exponential priors for both dimensions, which guides optimization towards a region with only poor performing high function values. 8a shows the prior and 8b shows optimization results after $D + 1 = 3$ initialization samples and 50 BO iterations. Note that, once again, optimization begins near the region incentivized by the prior, but moves away from the prior and towards the optima as BO progresses. After 50 BO iterations, PrBO finds all three optima regions of the Branin.

## C   Mathematical Derivations

### C.1   EI Derivation

Here, we provide a full derivation of Eq. (5):

$$EI_{f_\gamma}(\boldsymbol{x}) := \int_{-\inf}^{\inf} \max(f_\gamma - y, 0) p(y|\boldsymbol{x}) dy = \int_{-\inf}^{f_\gamma} (f_\gamma - y) \frac{p(\boldsymbol{x}|y)p(y)}{p(\boldsymbol{x})} dy. \tag{6}$$

As defined in Section 3.2, $p(y < f_\gamma) = \gamma$ and $\gamma$ is a quantile of the observed objective values $\{y^{(i)}\}$. Then $p(\boldsymbol{x}) = \int_{\mathbb{R}} p(\boldsymbol{x}|y)p(y) dy = \gamma g(\boldsymbol{x}) + (1-\gamma)b(\boldsymbol{x})$, where $g(\boldsymbol{x})$ and $b(\boldsymbol{x})$ are the posteriors introduced in Section 3.3. Therefore

$$\int_{-\inf}^{f_\gamma} (f_\gamma - y)p(\boldsymbol{x}|y)p(y) dy = g(\boldsymbol{x}) \int_{-\inf}^{f_\gamma} (f_\gamma - y)p(y) dy = \gamma f_\gamma g(\boldsymbol{x}) - g(\boldsymbol{x}) \int_{-\inf}^{f_\gamma} yp(y) dy, \tag{7}$$

so that finally

$$EI_{f_\gamma}(\boldsymbol{x}) = \frac{\gamma f_\gamma g(\boldsymbol{x}) - g(\boldsymbol{x}) \int_{-\inf}^{f_\gamma} yp(y) dy}{\gamma g(\boldsymbol{x}) + (1-\gamma)b(\boldsymbol{x})} \propto \left( \gamma + \frac{b(\boldsymbol{x})}{g(\boldsymbol{x})}(1-\gamma) \right)^{-1}. \tag{8}$$

### C.2   Proof of Proposition 1

Here, we provide the proof of Proposition 1:

$$
\begin{aligned}
\lim_{t \to \infty} \arg\max_{\boldsymbol{x} \in \mathcal{X}} EI_{f_\gamma}(\boldsymbol{x}) &= \lim_{t \to \infty} \arg\max_{\boldsymbol{x} \in \mathcal{X}} \int_{-\inf}^{f_\gamma} (f_\gamma - y)p(\boldsymbol{x}|y)p(y) dy & (9) \\[2mm]
&= \lim_{t \to \infty} \arg\max_{\boldsymbol{x} \in \mathcal{X}} g(\boldsymbol{x}) \int_{-\inf}^{f_\gamma} (f_\gamma - y)p(y) dy & (10) \\[2mm]
&= \lim_{t \to \infty} \arg\max_{\boldsymbol{x} \in \mathcal{X}} \left( \gamma f_\gamma g(\boldsymbol{x}) - g(\boldsymbol{x}) \int_{-\inf}^{f_\gamma} yp(y) dy \right) & (11) \\[2mm]
&= \lim_{t \to \infty} \arg\max_{\boldsymbol{x} \in \mathcal{X}} \frac{\gamma f_\gamma g(\boldsymbol{x}) - g(\boldsymbol{x}) \int_{-\inf}^{f_\gamma} yp(y) dy}{\gamma g(\boldsymbol{x}) + (1-\gamma)b(\boldsymbol{x})} & (12) \\[2mm]
&= \lim_{t \to \infty} \arg\max_{\boldsymbol{x} \in \mathcal{X}} \left( \gamma + \frac{b(\boldsymbol{x})}{g(\boldsymbol{x})}(1-\gamma) \right)^{-1} & (13) \\[2mm]
&= \lim_{t \to \infty} \arg\max_{\boldsymbol{x} \in \mathcal{X}} \left( \gamma + \frac{b(\boldsymbol{x})}{g(\boldsymbol{x})}(1-\gamma) \right)^{-\frac{1}{t}} & (14) \\[2mm]
&= \lim_{t \to \infty} \arg\max_{\boldsymbol{x} \in \mathcal{X}} \left( \gamma + \frac{P_b(\boldsymbol{x})\mathcal{M}_b(\boldsymbol{x})^{\frac{t}{\beta}}}{P_g(\boldsymbol{x})\mathcal{M}_g(\boldsymbol{x})^{\frac{t}{\beta}}}(1-\gamma) \right)^{-\frac{1}{t}} & (15) \\[2mm]
&= \lim_{t \to \infty} \arg\max_{\boldsymbol{x} \in \mathcal{X}} \left( \frac{P_b(\boldsymbol{x})\mathcal{M}_b(\boldsymbol{x})^{\frac{t}{\beta}}}{P_g(\boldsymbol{x})\mathcal{M}_g(\boldsymbol{x})^{\frac{t}{\beta}}}(1-\gamma) \right)^{-\frac{1}{t}} & (16) \\[2mm]
&= \lim_{t \to \infty} \arg\max_{\boldsymbol{x} \in \mathcal{X}} \left( \frac{P_b(\boldsymbol{x})}{P_g(\boldsymbol{x})} \right)^{-\frac{1}{t}} \left( \frac{\mathcal{M}_b(\boldsymbol{x})^{\frac{t}{\beta}}}{\mathcal{M}_g(\boldsymbol{x})^{\frac{t}{\beta}}} \right)^{-\frac{1}{t}} (1-\gamma)^{-\frac{1}{t}} & (17) \\[2mm]
&= \lim_{t \to \infty} \arg\max_{\boldsymbol{x} \in \mathcal{X}} \left( \frac{\mathcal{M}_b(\boldsymbol{x})}{\mathcal{M}_g(\boldsymbol{x})} \right)^{-\frac{1}{\beta}} & (18) \\[2mm]
&= \lim_{t \to \infty} \arg\max_{\boldsymbol{x} \in \mathcal{X}} \left( \frac{1 - \mathcal{M}_g(\boldsymbol{x})}{\mathcal{M}_g(\boldsymbol{x})} \right)^{-\frac{1}{\beta}} & (19)
\end{aligned}
$$

$$= \lim_{t \to \infty} \arg\max_{\boldsymbol{x} \in \mathcal{X}} \left( \frac{1}{\mathcal{M}_g(\boldsymbol{x})} - 1 \right)^{-\frac{1}{\beta}} \tag{20}$$

$$= \lim_{t \to \infty} \arg\max_{\boldsymbol{x} \in \mathcal{X}} (\mathcal{M}_g(\boldsymbol{x}))^{\frac{1}{\beta}} \tag{21}$$

$$= \lim_{t \to \infty} \arg\max_{\boldsymbol{x} \in \mathcal{X}} \mathcal{M}_g(\boldsymbol{x}) \tag{22}$$

$$\tag{23}$$

This shows that as iterations progress, the model grows more important. If PrBO is run long enough, the prior washes out and PrBO only trusts the probabilistic model.

## D  EXPERIMENTAL SETUP

We use a combination of publicly available implementations for our predictive models. For our Gaussian Process (GP) model, we use GPy's (GPy, since 2012) GP implementation with the Matérn5/2 kernel. We use different length-scales for each input dimensions, learned via Automatic Relevance Determination (ARD) (Neal, 2012). For our Random Forests (RF), we use scikit-learn's RF implementation (Pedregosa et al., 2011). We set the fraction of features per split to $0.5$, the minimum number of samples for a split to $5$ and disable bagging. We also adapt our RF implementation to use the same split selection approach as Hutter et al. (2014).

For our constrained Bayesian Optimization (cBO) approach, we use scikit-learn's RF classifier, trained on previously explored configurations, to predict the probability of a configuration being feasible. We then weight our EI acquisition function by this probability of feasibility, as proposed by Gardner et al. (2014). We normalize our EI acquisition function before considering the probability of feasibility, to ensure both values are in the same range. This cBO implementation is used in the Spatial use-case as in Nardi et al. (2019).

For all experiments, we set the model weight hyperparameter to $\beta = 10$ and the model quantile to $\gamma = 0.05$, see Appendices M and L. Before starting the main BO loop, PrBO is initialized by random sampling $D + 1$ points from the prior, where $D$ is the number of input variables. We use the public implementation of Spearmint[3], which by default uses 2 random samples for initialization. We set the bandwidth of our KDE priors to $100n^{-\frac{1}{D}}$, where $D$ is the number of input dimensions, see Appendix K. We normalize our KDE priors before computing the pseudo-posterior, to ensure they are in the same range as our model. We also implement interleaving which randomly samples a point to explore during BO with a $10\%$ chance.

We optimize our EI acquisition function using a multi-start local search, similar to the one used in SMAC (Hutter et al., 2011). Namely, we start local searches on the 10 best points evaluated in previous BO iterations, on the 10 best performing points from a set of $10,000$ random samples and on the 10 best performing points from $10,000$ random samples drawn from the prior. To compute the neighbors of each of these 30 total points, we normalize the range of each objective to $[0, 1]$ and randomly sample four neighbors from a truncated Gaussian centered at the original value and with standard deviation $\sigma = 0.2$.

We use four synthetic benchmarks in our experiments.

**Branin.** The Branin function is a well-known synthetic benchmark for optimization problems (Dixon, 1978). The Branin function has two input dimensions and three global minima.

**SVM.** is a hyperparameter-optimization benchmark in 2D based on Profet (Klein et al., 2019). This benchmark is generated by a generative meta-model built using a set of SVM classification models trained on 16 OpenML tasks. The benchmark has two input parameters, corresponding to SVM hyperparameters.

**FC-Net.** is a hyperparameter and architecture optimization benchmark in 6D based on Profet. The FC-Net benchmark is generated by a generative meta-model built using a set of feed-forward neural networks trained on the same 16 OpenML tasks as the SVM benchmark. The benchmark has six input parameters corresponding to network hyperparameters.

---

[3]https://github.com/HIPS/Spearmint

Table 1: Search spaces for our synthetic benchmarks. For the Profet benchmarks, we report the original ranges and whether or not a log scale was used.

| Benchmark | Parameter name | Parameter values | Log scale |
|---|---|---|---|
| Branin | $x_1$ | $[-5, 10]$ | - |
|  | $x_2$ | $[0, 15]$ | - |
| SVM | C | $[e^{-10}, e^{10}]$ | ✓ |
|  | $\gamma$ | $[e^{-10}, e^{10}]$ | ✓ |
| FCNet | learning rate | $[10^{-6}, 10^{-1}]$ | ✓ |
|  | batch size | $[8, 128]$ | ✓ |
|  | units layer 1 | $[16, 512]$ | ✓ |
|  | units layer 2 | $[16, 512]$ | ✓ |
|  | dropout rate l1 | $[0.0, 0.99]$ | - |
|  | dropout rate l2 | $[0.0, 0.99]$ | - |
| XGBoost | learning rate | $[10^{-6}, 10^{-1}]$ | ✓ |
|  | gamma | $[0, 2]$ | - |
|  | L1 regularization | $[10^{-5}, 10^3]$ | ✓ |
|  | L2 regularization | $[10^{-5}, 10^3]$ | ✓ |
|  | number of estimators | $[10, 500]$ | - |
|  | subsampling | $[0.1, 1]$ | - |
|  | maximum depth | $[1, 15]$ | - |
|  | minimum child weight | $[0, 20]$ | - |

**XGBoost.** is hyperparameter-optimization benchmark in 8D based on Profet. The XGBoost benchmark is generated by a generative meta-model built using a set of XGBoost regression models in 11 UCI datasets. The benchmark has eight input parameters, corresponding to XGBoost hyperparameters.

The search spaces for each benchmark are summarized in Table 1. For the Profet benchmarks, we report the original ranges and whether or not a log scale was used. However, in practice, Profet's generative model transforms the range of all hyperparameters to a linear $[0, 1]$ range. We use Emukit's public implementation for these benchmarks (Paleyes et al., 2019).

## E  SVM REGRET COMPARISON

In addition to the experiments in Section 4.3, we show the performance of PrBO on the SVM benchmark. Figure 9 shows a log regret comparison of PrBO, Spearmint, Prior Sampling and $10{,}000\times$ RS. We note that the results are similar to the other benchmarks in Figure 4. Namely, PrBO with a strong prior outperforms RS and spearmint. PrBO also outperforms Spearmint with a weak prior and even with a uniform prior.

## F  SPATIAL REAL-WORLD APPLICATION

`Spatial` (Koeplinger et al., 2018) is a programming language and corresponding compiler for the design of application accelerators on reconfigurable architectures, e.g. field-programmable gate arrays (FPGAs). These reconfigurable architectures are a type of logic chip that can be reconfigured via software to implement different applications. `Spatial` provides users with a high-level of abstraction for hardware design, so that they can easily design their own applications on FPGAs. It allows users to specify parameters that do not change the behavior of the application, but impact the runtime and resource-usage (e.g. logic units) of the final design. During compilation, the `Spatial` compiler estimates the ranges of these parameters and estimates the resource-usage and runtime of the application for different parameter values. These parameters can then be optimized during compilation in order to achieve the design with the fastest runtime. We fully integrate PrBO as a pass in `Spatial`'s compiler, so that `Spatial` can automatically use PrBO for the optimization during compilation. This enables `Spatial` to seamlessly call PrBO during the compilation of any new application to guide the search towards the best design on an application-specific basis.

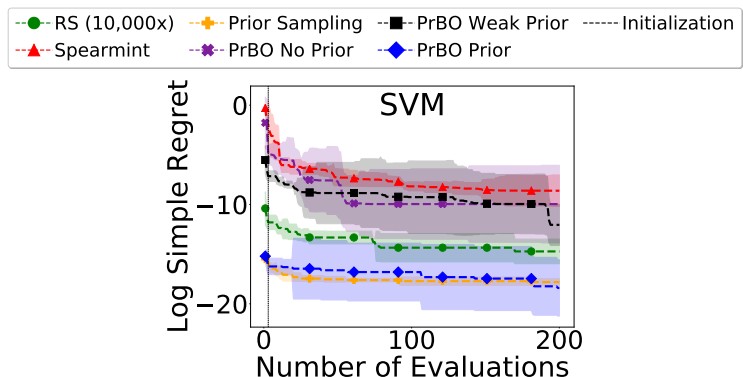

Figure 9: Log regret comparison of PrBO with and without priors, $10,000\times$ RS, and Spearmint (mean +/- one std on 5 repetitions). We run the benchmark for 200 iterations.

Table 2: Search space, priors, and expert configuration for the MD Grid application. The default value for each parameter is shown in bold.

| Parameter name | Type | Values | Expert | Prior |
|---|---|---|---|---|
| loop_grid0_z | Ordinal | [**1**, 2, ..., 15, 16] | 1 | [0.2, 0.1, 0.05, 0.05, 0.05, 0.05, 0.05, 0.05, 0.05, 0.05, 0.05, 0.05, 0.05, 0.05, 0.05, 0.05] |
| loop_q | Ordinal | [**1**, 2, ..., 31, 32] | 8 | [0.08, 0.08, 0.02, 0.1, 0.02, 0.02, 0.02, 0.1, 0.02, 0.02, 0.02, 0.02, 0.02, 0.02, 0.02, 0.1, 0.02, 0.02, 0.02, 0.02, 0.02, 0.02, 0.02, 0.02, 0.02, 0.02, 0.02, 0.02, 0.02, 0.02, 0.02, 0.02] |
| par_load | Ordinal | [**1**, 2, 4] | 1 | [0.45, 0.1, 0.45] |
| loop_p | Ordinal | [**1**, 2, ..., 31, 32] | 2 | [0.1, 0.1, 0.1, 0.1, 0.05, 0.03, 0.02, 0.02, 0.02, 0.02, 0.02, 0.02, 0.02, 0.02, 0.02, 0.02, 0.02, 0.02, 0.02, 0.02, 0.02, 0.02, 0.02, 0.02, 0.02, 0.02, 0.02, 0.02, 0.02, 0.02, 0.02, 0.02] |
| loop_grid0_x | Ordinal | [**1**, 2, ..., 15, 16] | 1 | [0.2, 0.1, 0.05, 0.05, 0.05, 0.05, 0.05, 0.05, 0.05, 0.05, 0.05, 0.05, 0.05, 0.05, 0.05, 0.05] |
| loop_grid1_z | Ordinal | [**1**, 2, ..., 15, 16] | 1 | [0.2, 0.2, 0.1, 0.1, 0.07, 0.03, 0.03, 0.03, 0.03, 0.03, 0.03, 0.03, 0.03, 0.03, 0.03, 0.03] |
| loop_grid0_y | Ordinal | [**1**, 2, ..., 15, 16] | 1 | [0.2, 0.1, 0.05, 0.05, 0.05, 0.05, 0.05, 0.05, 0.05, 0.05, 0.05, 0.05, 0.05, 0.05, 0.05, 0.05] |
| ATOM1LOOP | Categorical | [false, **true**] | true | [0.1, 0.9] |
| ATOM2LOOP | Categorical | [false, **true**] | true | [0.1, 0.9] |
| PLOOP | Categorical | [false, **true**] | true | [0.1, 0.9] |

In our experiments, we introduce for the first time the automatic optimization of three `Spatial` real-world applications, namely, 7D shallow and deep CNNs, and a 10D molecular dynamics grid application. Previous work by Nardi et al. (2019) had applied automatic optimization of `Spatial` parameters on a set of benchmarks but in our work we focus on real-world applications raising the bar of state-of-the-art automated hardware design optimization. PrBO is used to optimize the parameters to find a design that leads to the fastest runtime. The search space for these three applications is based on ordinal and categorical parameters; to handle these discrete parameters in the best way we implement and use a Random Forest surrogate instead of a Gaussian Process one as explained in Appendix D. These parameters are application specific and control how much of the FPGAs' resources we want to use to parallelize each step of the application's computation. The goal here is to find which steps are more important to parallelize in the final design, in order to achieve the fastest

Table 3: Search space, priors, and expert configuration for the Shallow CNN application. The default value for each parameter is shown in bold.

| Parameter name | Type | Values | Expert | Prior |
|---|---|---|---|---|
| LP | Ordinal | [**1**, 4, 8, 16, 32] | 16 | [0.4, 0.065, 0.07, 0.065, 0.4] |
| P1 | Ordinal | [**1**, 2, 3, 4] | 1 | [0.1, 0.3, 0.3, 0.3] |
| SP | Ordinal | [**1**, 4, 8, 16, 32] | 16 | [0.4, 0.065, 0.07, 0.065, 0.4] |
| P2 | Ordinal | [**1**, 2, 3, 4] | 4 | [0.1, 0.3, 0.3, 0.3] |
| P3 | Ordinal | [**1**, 2, ..., 31, 32] | 1 | [0.1, 0.1, 0.033, 0.1, 0.021, 0.021, 0.021, 0.1, 0.021, 0.021, 0.021, 0.021, 0.021, 0.021, 0.021, 0.021, 0.021, 0.021, 0.021, 0.021, 0.021, 0.021, 0.021, 0.021, 0.021, 0.021, 0.021, 0.021, 0.021, 0.021, 0.021, 0.021] |
| P4 | Ordinal | [**1**, 2, ..., 47, 48] | 4 | [0.08, 0.0809, 0.0137, 0.1, 0.0137, 0.0137, 0.0137, 0.1, 0.0137, 0.0137, 0.0137, 0.05, 0.0137, 0.0137, 0.0137, 0.0137, 0.0137, 0.0137, 0.0137, 0.0137, 0.0137, 0.0137, 0.0137, 0.0137, 0.0137, 0.0137, 0.0137, 0.0137, 0.0137, 0.0137, 0.0137, 0.0137, 0.0137, 0.0137, 0.0137, 0.0137, 0.0137, 0.0137, 0.0137, 0.0137, 0.0137, 0.0137, 0.0137, 0.0137, 0.0137, 0.0137, 0.0137, 0.0137] |
| x276 | Categorical | [false, **true**] | true | [0.1, 0.9] |

Table 4: Search space, priors, and expert configuration for the Deep CNN application. The default value for each parameter is shown in bold.

| Parameter name | Type | Values | Expert | Prior |
|---|---|---|---|---|
| LP | Ordinal | [**1**, 4, 8, 16, 32] | 8 | [0.4, 0.065, 0.07, 0.065, 0.4] |
| P1 | Ordinal | [**1**, 2, 3, 4] | 1 | [0.4, 0.3, 0.2, 0.1] |
| SP | Ordinal | [**1**, 4, 8, 16, 32] | 8 | [0.4, 0.065, 0.07, 0.065, 0.4] |
| P2 | Ordinal | [**1**, 2, 3, 4] | 2 | [0.4, 0.3, 0.2, 0.1] |
| P3 | Ordinal | [**1**, 2, ..., 31, 32] | 1 | [0.04, 0.01, 0.01, 0.1, 0.01, 0.01, 0.01, 0.1, 0.01, 0.01, 0.01, 0.01, 0.01, 0.01, 0.01, 0.2, 0.01, 0.01, 0.01, 0.01, 0.01, 0.01, 0.1, 0.01, 0.01, 0.01, 0.01, 0.01, 0.01, 0.01, 0.2] |
| P4 | Ordinal | [**1**, 2, ..., 47, 48] | 4 | [0.05, 0.005, 0.005, 0.005, 0.005, 0.005, 0.005, 0.13, 0.005, 0.005, 0.005, 0.005, 0.005, 0.005, 0.005, 0.2, 0.005, 0.005, 0.005, 0.005, 0.005, 0.005, 0.005, 0.11, 0.005, 0.005, 0.005, 0.005, 0.005, 0.005, 0.2, 0.005, 0.005, 0.005, 0.005, 0.005, 0.005, 0.005, 0.005, 0.005, 0.005, 0.005, 0.005, 0.005, 0.005, 0.005, 0.005, 0.1] |
| x276 | Categorical | [false, **true**] | true | [0.1, 0.9] |

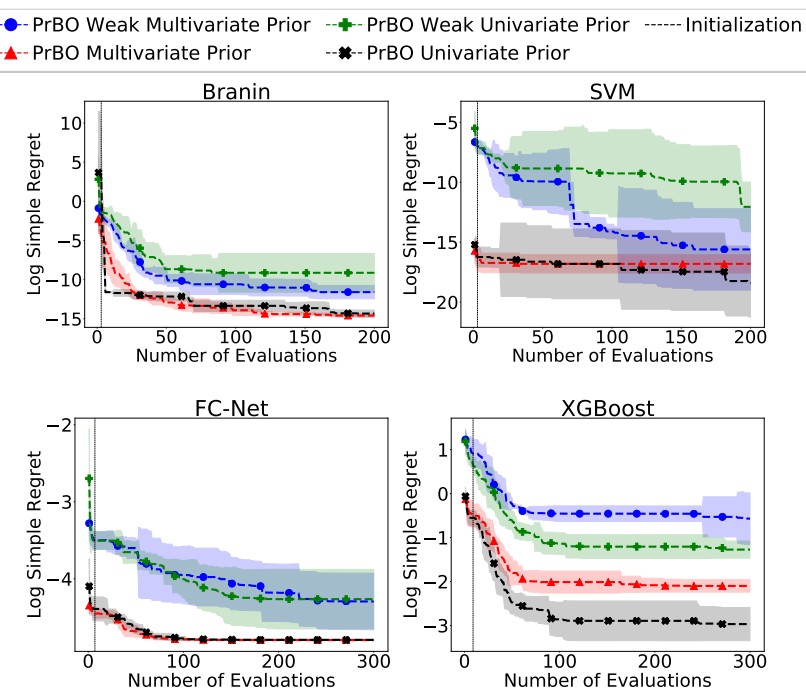

Figure 10: Log regret comparison of PrBO with multivariate and univariate KDE priors. The line and shaded regions show the mean and standard deviation of the log simple regret after 5 runs. All methods were initialized with $D + 1$ random samples, where $D$ is the number of input dimensions, indicated by the vertical dashed line. We run the benchmarks for $100D$ iterations, capped at 300.

runtime. Some parameters also control whether we want to enable pipeline scheduling or not, which consumes resources but accelerates runtime. We refer to Koeplinger et al. (2018) and Nardi et al. (2019) for more details on Spatial's parameters.

The three Spatial benchmarks also have feasibility constraints in the search space, meaning that some parameter configurations are infeasible. A configuration is considered infeasible if the final design requires more logic resources than what the FPGA provides, i.e., it is not possible to perform FPGA synthesis because the design does not fit in the FPGA. To handle these constraints, we use our cBO implementation (Appendix D). Our goal is thus to find the design with the fastest runtime under the constraint that the design fits the FPGA resource budget.

The priors for these Spatial applications take the form of a list of probabilities, containing the probability of each ordinal or categorical value being good. Each benchmark also has a default configuration, which ensures all methods start with at least one feasible configuration. The priors and the default configuration for these benchmarks were provided once by an unbiased Spatial developer, who is not an author of this paper, and kept unchanged during the entire project. The search space, priors, and the expert configuration used in our experiments for each application are presented in Tables 2, 3, and 4.

## G    MULTIVARIATE PRIOR COMPARISON

Figure 10 shows a log regret comparison of PrBO with univariate and multivariate KDE priors. We show results for univariate and multivariate versions of our weak and strong KDE priors. We use the best $10D$ points out of $1,000D$ and $10,000,000D$ randomly sampled points to create our weak and strong priors, respectively. We use the same points to create the univariate and multivariate priors. We recall that the goal of these synthetic priors is to have an unbiased prior for our experiments, whereas manual priors would be biased by our own expertise of these benchmarks. In practice, users will manually define these priors without needing additional experiments.

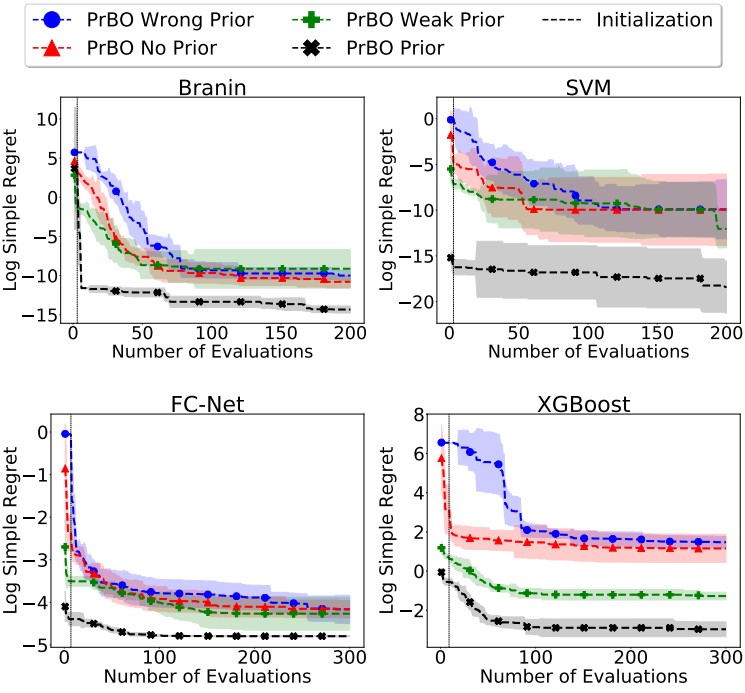

Figure 11: Log regret comparison of PrBO with varying prior quality. The line and shaded regions show the mean and standard deviation of the log simple regret after 5 runs. All methods were initialized with $D + 1$ random samples, where $D$ is the number of input dimensions, indicated by the vertical dashed line. We run the benchmarks for $100D$ iterations, capped at 300.

We note that in all cases PrBO achieves similar performance with univariate and multivariate priors. For the Branin and SVM benchmarks, the weak multivariate prior leads to slightly better results than the weak univariate prior. However, we note that the difference is small, in the order of $10^{-4}$ and $10^{-6}$, respectively.

Surprisingly, for the XGBoost benchmark, the univariate version for both the weak and strong priors lead to better results than their respective multivariate counterparts, though, once again, the difference in performance is small, around $0.2$ and $0.03$ for the weak and strong prior, respectively, whereas the XGBoost benchmark can reach values as high as $f(\boldsymbol{x}) = 600$. Our hypothesis is that this difference comes from the bandwidth estimator ($100n^{-\frac{1}{D}}$), which leads to larger bandwidths, consequently, smoother priors, when a multivariate prior is constructed.

## H  MISLEADING PRIOR COMPARISON

Figure 11 shows the effect of injecting a misleading prior in PrBO. We compare PrBO with a misleading prior, no prior, a weak prior, and a strong prior. To create the misleading prior, we use a univariate KDE prior built with the worst $10D$ out of $10,000,000D$ random samples. For all benchmarks, we note that the misleading prior slows down convergence, as expected, since it pushes the optimization away from the optima in the initial phase. However, PrBO is still able to forget the misleading prior and achieve similar regret to PrBO without a prior.

## I  COMPARISON TO OTHER BASELINES

We compare PrBO to SMAC (Hutter et al., 2011) and TPE (Bergstra et al., 2011) on our four synthetic benchmarks. We use Hyperopt's implementation[4] of TPE and the SMAC3 Python implementation

---

[4]https://github.com/hyperopt/hyperopt

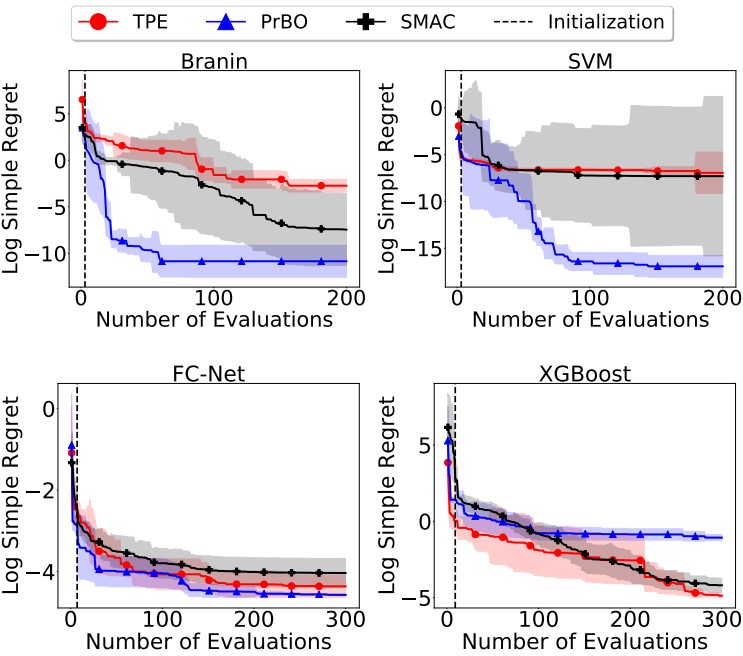

Figure 12: Log regret comparison of PrBO, SMAC, and TPE. The line and shaded regions show the mean and standard deviation of the log simple regret after 5 runs. PrBO was initialized with $D + 1$ random samples, where $D$ is the number of input dimensions, indicated by the vertical dashed line. We run the benchmarks for $100D$ iterations, capped at 300.

of SMAC[5]. Hyperopt defines priors as one of a list of supported distributions, including Uniform, Normal, and Lognormal distributions. Since it is not possible to input the KDE priors introduced in Section 4.1 into the TPE algorithm, we instead use manually defined priors in the format supported by the HyperOpt implementation. We note that this is straightforward in PrBO, as PrBO supports any form of probability distribution as a prior. We are then able to perform a fair comparison between the two approaches that use the same exact prior. Since SMAC does not support probability distribution priors, we do not inject any priors in SMAC.

We define the prior for each input parameter as a Gaussian distribution with mean at the optimum and with standard deviation equal to half of the parameters range. For the Branin prior, we arbitrarily choose one of the optima, i.e., the $(\pi, 2.275)$ optimum. For the Profet benchmarks, we use the minimum out of $10,000,000D$ random samples as an approximation of the optimum. We note that using Hyperopt's Gaussian priors leads to an unbounded search space, which sometimes leads TPE to suggest parameter configurations outside the allowed parameter range. To prevent these values from being evaluated, we convert values outside the parameter range to be equal to the upper or lower range limit, depending on which limit was exceeded.

Figure 12 shows a log regret comparison between PrBO, SMAC, and TPE on our four synthetic benchmarks. PrBO outperforms SMAC on three out of four benchmarks, and shows consistent speedups in the early phases of the optimization proces. PrBO also outperforms TPE in three out of four benchmarks, namely, Branin, SVM, and FCNet. We note, however, that the good performance of TPE on XGBoost may be an artifact of the approach of clipping values to its maximal or minimal values as mentioned above. In fact, the clipping nudges TPE towards promising configurations in this case, since XGBoost has low function value near the edges of the search space. Overall, the better performance of PrBO is expected, since PrBO is able to combine prior knowledge with more sample-efficient surrogates, which leads to better performance.

---

[5]https://github.com/automl/SMAC3

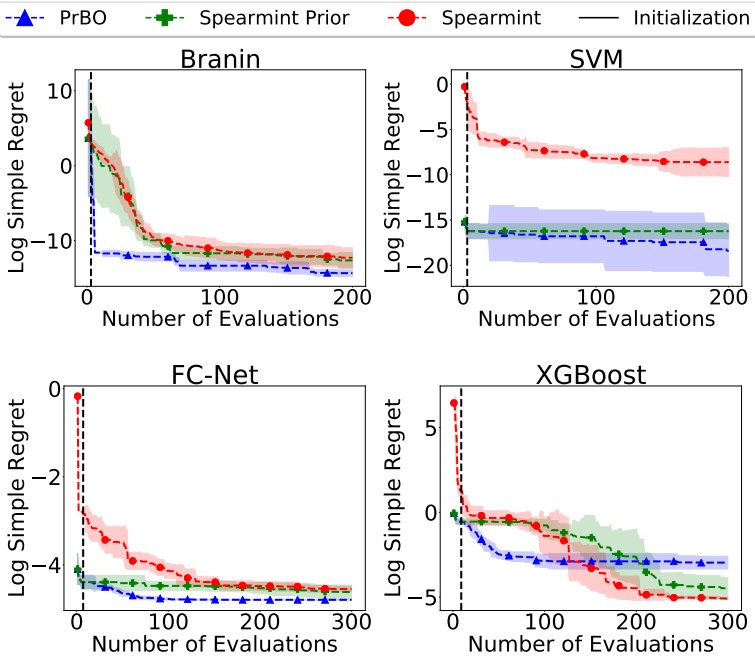

Figure 13: Log regret comparison of PrBO, Spearmint with prior initialization, and Spearmint with default initialization. The line and shaded regions show the mean and standard deviation of the log simple regret after 5 runs. PrBO and Spearmint Prior were initialized with $D + 1$ random samples from the prior, where $D$ is the number of input dimensions, indicated by the vertical dashed line. We run the benchmarks for $100D$ iterations, capped at 300.

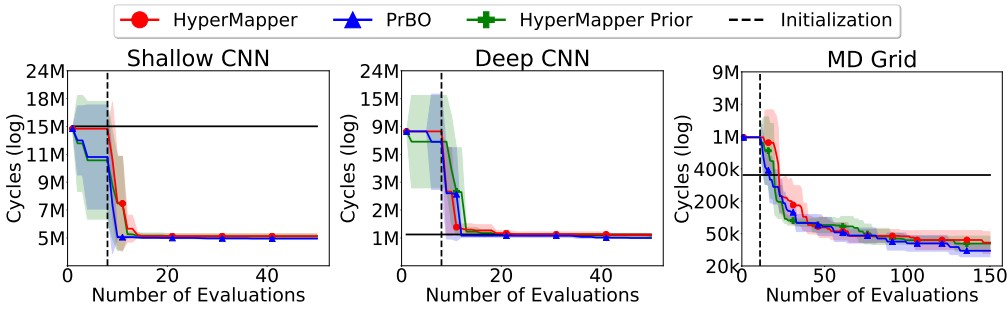

Figure 14: Log regret comparison of PrBO, HyperMapper with prior initialization, and HyperMapper with default initialization. The line and shaded regions show the mean and standard deviation of the log simple regret after 5 runs. PrBO and HyperMapper Prior were initialized with $D + 1$ random samples from the prior, where $D$ is the number of input dimensions, indicated by the vertical dashed line.

## J  PRIOR BASELINES COMPARISON

In this section, we compare PrBO with Spearmint and HyperMapper using the same prior initialization on all of our benchmarks (Figs. 13 and 14). Both PrBO and the baselines were initialized with the same $D + 1$ samples from the prior, indicated by the vertical dashed line. We run each optimizer for 5 times and report mean and standard deviation of the 5 runs. For comparison, we also show the baselines with the default initialization.

Figure 13 shows the comparison between PrBO and Spearmint Prior. In most benchmarks, the prior initialization does not lead to improved final performance. In particular, for XGBoost, the prior leads

to improvement in early iterations, but to worse final performance. We note that for SVM, the prior leads to better performance, however, we note that the improved performance is given solely from sampling form the prior. There is no improvement for Spearmint Prior after initialization. In contrast, in all cases, PrBO is able to leverage the prior both during initialization and its Bayesian Optimization phase, leading to improved performance. PrBO still outperforms Spearmint Prior in the same 3 out of 4 benchmarks.

Figure 14 shows similar results for our Spatial benchmarks. Once again, the prior does not lead HyperMapper to improved final performance. For the Shallow CNN benchmark, the prior leads HyperMapper to improved performance in early iterations, compared to HyperMapper with default initialization, but HyperMapper Prior is still outperformed by PrBO. Additionaly, the prior leads to degraded performance in the Deep CNN benchmark. These results confirm that PrBO is able to leverage the prior in its pseudo-posterior during optimization, leading to improved performance in almost all benchmarks compared to state-of-the-art BO baselines.

## K    PRIOR BANDWIDTH SELECTION

We show the effect of different bandwidth sizes on the univariate KDE prior. For that, we compare the performance of sampling from the prior and PrBO with different bandwidth sizes. We use scipy's Gaussian KDE implementation and modify its bandwidth size with four variations of Scott's Rule $an^{-\frac{1}{D+b}}$. We experiment with $a = 1$, $b = 4$ (scipy's default); $a = 1$, $b = 0$; $a = 10$, $b = 0$; and $a = 100$, $b = 0$. Note that larger values for $a$ and smaller values for $b$ lead to smaller bandwidths. For each bandwidth size, we show results for an array of varying quality priors. We select a constant $10D$ points in each prior and vary the size of the uniform random sample dataset. We follow the following rule: we use the best performing $10D$ samples to create the prior from a random sample dataset size of $10D\frac{100}{x}$; we refer to this prior as $x\%$. We experiment with dataset sizes varying from $10D$ to $10^7 D$.

Figure 15 shows the performance of purely sampling from the prior. We note that, in most cases, using a larger dataset leads to better results. This is expected, sampling more points means we find more points near the optima and, therefore, the prior will be built with points closer to the optima. Likewise, we note that smaller bandwidths often lead to better results, especially as more points are sampled. This is also expected, since a smaller bandwidth means the prior distribution will be more peaked around the optima. However, there are a couple of exceptions to these trends. First, for the Branin, sampling more points does not lead to a better prior when we use $a = 1, b = 4$, this is likely because the multiple minima of the Branin and the bigger bandwidth lead the prior to be oversmoothed, missing the peaks near the optima. Second, smaller bandwidths do not always lead to better performance for smaller random sample datasets. This happens because we find points farther from the optima in these datasets and end up computing priors peaked at points that are farther from the optima, i.e., our priors become misleading. The effects of these misleading priors can be especially noticed for the 100% random samples dataset. Based on these results, we set our KDE priors to $100n^{-\frac{1}{D}}$, where $D$ is the number of input dimensions.

Figure 16 shows the performance of PrBO for different priors. The same observations from Figure 15 hold here. Namely, sampling more points and using smaller bandwidths lead to better performance. Also, the 100% dataset once again leads to inconsistent results, since it is a misleading prior for PrBO. Based on these results, we use the smallest bandwidth and largest dataset in our experiments, i.e. $a = 100, b = 0$, and 0.0001%. Intuitively, this is a reasonable choice, since these priors will be our closest approximation to an ideal prior that is centered exactly at the optima, where sampling from the prior always leads to the optimum. Our results in Figures 15 and 16 shows that this combination leads to the best results in all benchmarks, as expected.

Figure 17 shows a performance comparison between PrBO and sampling from the prior. For these results, we use $a = 100, b = 0$ and compare the regret of PrBO and sampling from the prior for different dataset sizes. PrBO performs better for nearly all dataset sizes and benchmarks. This is expected as PrBO complements the prior with its probabilistic model, learning which regions within the prior are better to explore and also recovering from misleading priors. There are two exceptions on the SVM benchmark, where sampling from the prior performs slightly better for 0.01% and

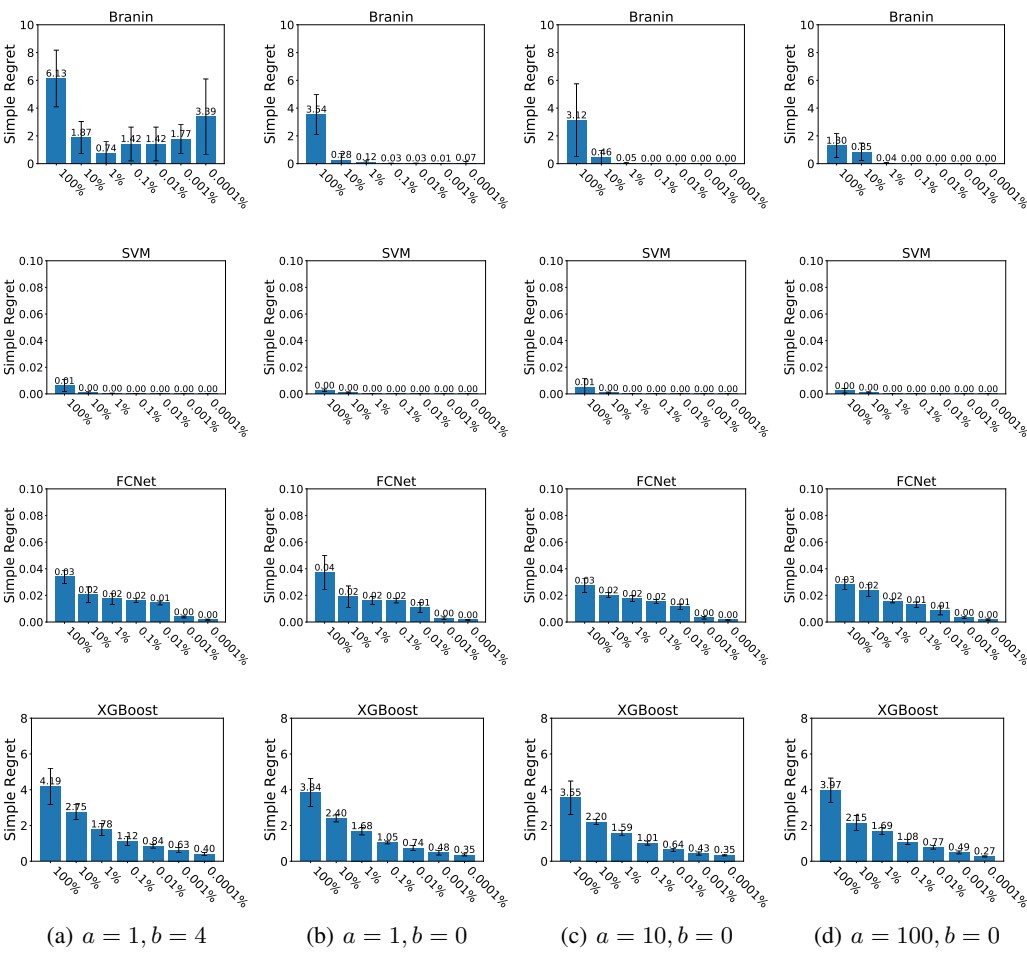

Figure 15: Simple regret of sampling from the prior with different priors for our synthetic benchmarks. We provide 5 repetitions for each experiment and mean +/- one std error bars. A more informative prior gives better results in all benchmarks.

0.0001% datasets. We note, however, that the difference in performance is extremely small, in the order of $10^{-5}$ and $10^{-7}$, respectively.

# L  $\gamma$-SENSITIVITY STUDY

We show the effect of the $\gamma$ hyperparameter introduced in Section 3.2 for the quantile identifying the points considered to be good. To show this, we compare the performance of PrBO with a weak KDE prior and different $\gamma$ values. We use our weak prior as it leads to greater variation in performance, which helps to visualize better the impact of the $\gamma$ hyperparameter. For all experiments, we initialize PrBO with $D + 1$ random samples and then run PrBO until it reaches $10D$ function evaluations. For each $\gamma$ value, we run PrBO five times and report mean and standard deviation.

Figure 18 shows the results of our comparison. We first note that values near the lower and higher extremes lead to degraded performance, this is expected, since these values will lead to an excess of either exploitation or exploration. Further, we note that PrBO achieves similar performance for all values of $\gamma$, however, $\gamma = 0.03$ and $\gamma = 0.05$ consistently lead to better performance, with $\gamma = 0.05$ usually leading to lower deviation.

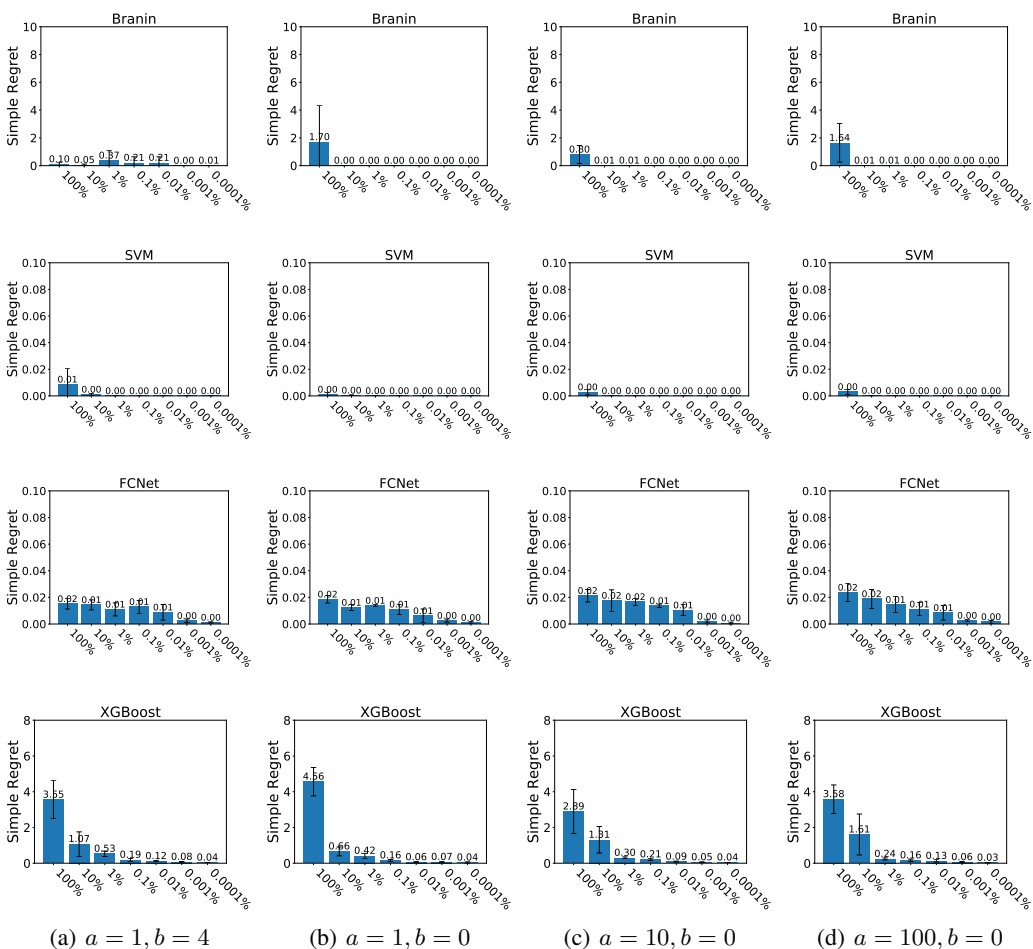

Figure 16: Simple regret of PrBO with different priors for our synthetic benchmarks. We provide 5 repetitions for each experiment and mean +/- one std error bars. A more informative prior gives better results in all benchmarks.

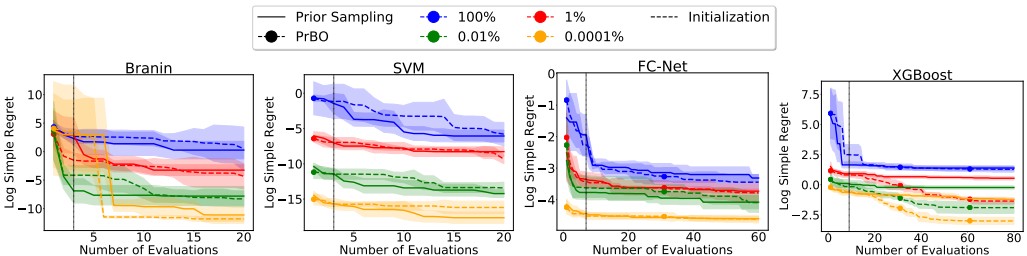

Figure 17: Log simple regret comparison between PrBO and sampling from the prior. The shaded lines are mean +/- one std error. PrBO was initialized with $D + 1$ random samples, indicated by the vertical dashed line.

# M $\beta$-SENSITIVITY STUDY

We show the effect of the $\beta$ hyperparameter introduced in Section 3.3 for controlling the influence of the prior over time. To show the effects of $\beta$, we compare the performance of PrBO with a weak KDE prior and different $\beta$ values on our four synthetic benchmarks. We use our weak prior as it leads to

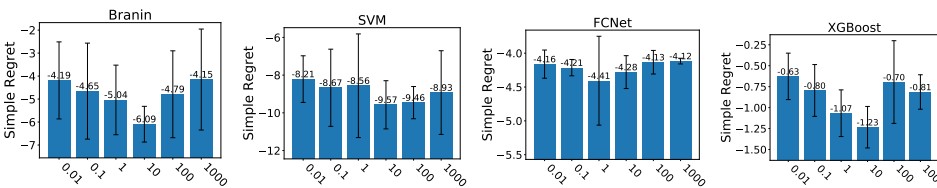

Figure 18: Comparison of PrBO with a weak KDE prior and different values for the $\gamma$ hyperparameter on our four synthetic benchmarks. We run PrBO with a budget of $10D$ function evaluations, including $D + 1$ randomly sampled DoE configurations.

Figure 19: Comparison of PrBO with a weak KDE prior and different values for the $\beta$ hyperparameter on our four synthetic benchmarks. We run PrBO with a budget of $10D$ function evaluations, including $D + 1$ randomly sampled DoE configurations.

greater variation in performance, which helps to visualize better the impact of the $\beta$ hyperparameter. For all experiments, we initialize PrBO with $D + 1$ random samples and then run PrBO until it reaches $10D$ function evaluations. For each $\beta$ value, we run PrBO five times and report mean and standard deviation.

Figure 19 shows the results of our comparison. We note that values of $\beta$ that are too low (near 0.01) or too high (near 1000) lead to lower performance. This shows that putting too much emphasis on the model or the prior will lead to degraded performance, as expected. Further, we note that $\beta = 10$ lead to the best performance in three out of our four benchmarks. This result is reasonable, as $\beta = 10$ means PrBO will put more emphasis on the prior in early iterations, when the predictive model is still not accurate, and slowly shift towards putting more emphasis on the model as the model sees more data and becomes more accurate.

