# OpenReview forum: "Prior-guided Bayesian Optimization"
_ICLR.cc/2021/Conference — Reject_

### Official Review · AnonReviewer2 · 2020-10-26
**Review of Prior-guided Bayesian Optimization**

**Rating:** 6
**Confidence:** 5

**Review:**

The goal of this paper is to enable the introduction of prior expert knowledge in Bayesian optimization. This is performed by defining a prior distribution for the optimal value, which is included in the pseudo-posterior used to select new points by Expected Improvement. The number of iterations it takes to overcome a potentially wrong prior information is controlled by a parameter, whose sensitivity is studied. Extensive experiments are conducted on toy examples as well are more realistic hyper-parameter test cases.

This paper could interest practitioners, but some aspects need to be better detailed and more comparison to existing methods should be added.

Detailed comments:

The proposed model in (3) amount to scale the probability of improvement with an arbitrary threshold by a probability of optimality given by the prior. This is very similar to the use of the probability of feasibility in constrained BO, see, e.g., Schonlau, M.; Welch, W. J. & Jones, D. R. Global versus local search in constrained optimization of computer models. Lecture Notes-Monograph Series, JSTOR, 1998, 11-25 and references therein. Hence couldn’t this so called pseudo-posterior be used directly?

There are additional references tackling the introduction of a prior (or lack of) to compare with, both in the state of the art and experiments, e.g.,:
- Anil Ramachandran, Sunil Gupta, Santu Rana, Cheng Li, Svetha Venkatesh, Incorporating expert prior in Bayesian optimisation via space warping, Knowledge-Based Systems, Volume 195, 2020.
- Bobak Shahriari, Alexandre Bouchard-Cote, Nando Freitas ; Proceedings of the 19th International Conference on Artificial Intelligence and Statistics, PMLR 51:1168-1176, 2016.

Another naive competitor is the use of the prior on the optimum only at the design of experiments stage, keeping the usual BO loop. This baseline should be added too.

Minor details:

Figure 1: consider showing the density model given by (4) here too, not just in appendix

Experiments: what is exactly the weak prior setting?

P8: on Oh et al., this is a structural prior to cope with high-dimensional optimization, so not clearly related.

---

> ### Author Response · Authors · 2020-11-14
> **Response to AnonReviewer2**
>
> We are glad that the reviewer found our experiments extensive and interesting. We thank you for the constructive feedback and address the main concerns raised below.
>
> **Q1.** This is very similar to the use of the probability of feasibility in constrained BO. Hence couldn’t this so-called pseudo-posterior be used directly?
> **A1.** This is an interesting proposal. One issue would be that the probability of feasibility is updated over time, whereas the user prior is not. Using a fixed prior in the same way as the probability of feasibility would thus not lead to the prior being washed out and would not be robust under misspecified priors. But combining it with our approach of down-weighting the prior over time would indeed be an alternative approach for future work; thank you for the suggestion!
>
> **Q2.** There are additional references tackling the introduction of a prior to compare
> **A2.** Our work is conceptually different from the work of Shahriari et al. (2016). Their work focuses on unbounded search spaces. The priors in their work are not about good regions of the space, but rather a regularization function that penalizes configurations based on their distance to the center of the user-defined search space. Users in their work may not even be aware of the priors that are used.
> The work of Ramachandran et al. (2020) is more closely related to PrBO, however, PrBO sets itself apart in 3 ways:
> 1. PrBO is agnostic to the probabilistic model used, while their approach is restricted to GPs. Again, this is, e.g., important in our practical applications with many discrete variables, where RFs work much better.
> 2. PrBO explicitly controls the balance between prior and model via the beta hyperparameter.
> 3. PrBO’s probabilistic model is fitted independently from the prior, which ensures it is not biased by the prior. Their approach fits the model to a warped version of the space, transformed using the prior, making it more difficult to recover from misleading priors.
>
> We note that Ramachandran et al. do not provide a public implementation of their work and the paper includes several underspecified parts leaving us unable to reimplement it for a comparison. The paper also does not compare to any state-of-the-art approaches such as Spearmint, TPE, or SMAC, and it appears to perform better with worse priors than with good ones (compare their Figure 5c vs Figure 5a), which makes it unclear how well the approach works. We added both works to our Related Work section.
>
> **Q3.** Another naive competitor is the use of the prior on the optimum only at the design of experiments stage, keeping the usual BO loop. This baseline should be added too.
> **A3.** Thanks, we in fact already compared to this baseline in Appendix J.
>
> **Q4.** Experiments: what is exactly the weak prior setting?
> **A4.** Sorry, this is a KDE fit on the best 10D out of 1,000D uniformly sampled points. We clarified this in Section 4.3.
>
> **Q5.** P8: on Oh et al., this is a structural prior to cope with high-dimensional optimization, so not clearly related.
> **A5.** Thanks, we made this distinction clear in our related works.

---

> > ### Comment · AnonReviewer2 · 2020-11-23
> > **Author's response**
> >
> > Thank you for the response, it clarifies some technical aspects. As for the novelty and impact, it remains a subject of discussion.
> >
> > A1: a similar decay rate as used here would work.

---

### Official Review · AnonReviewer3 · 2020-10-27
**Interesting Algorithm but Questionable Usefulness and Novelty**

**Rating:** 4
**Confidence:** 4

**Review:**

Summary:
This paper incorporates a prior distribution given by experts into Bayesian optimization (BO), to leverage useful human knowledge to accelerate BO. The algorithm uses an intuitive approach to combine the prior with the probabilistic surrogate model of BO to derive a pseudo-posterior, which naturally leads the EI acquisition function. As the BO progresses, the prior information is gradually overwhelmed by the observed data, which ensures the asymptotically correct behaviour.

Strong points:
- The design of the algorithm is intuitive and natural.
- Experiments are comprehensive. Figure 2 is a particularly nice illustration.
- The paper is well written.

Weak points/questions:
- I'm not fully convinced whether it's really useful to ask an expert to specify a prior distribution and then run the proposed algorithm, compared with the baseline of simply asking the expert to specify the search space and then running standard BO. Firstly, asking the expert to specify a search space is intuitively easier for the expert, compared with asking them to give a distribution across the entire domain. Moreover, I think it's hard to justify why the proposed approach is better in practice than simply asking the expert for a reduced search space. I would like to see either a comparison with this baseline (preferably for the experiment in Section 4.4 if possible), or a justification regarding why this baseline isn't competitive.
- The proposed algorithm is a straightforward combination of PI, TPE, and EI. Therefore, the technical contribution may not be novel enough.
- Figure 4: It seems that Spearmint consistently outperforms PrBO with weak prior. Does this mean that if you don't have a good prior, you should simply run standard BO?
- There is only one real-world experiment (Section 4.4) where an expert is asked to specify a prior, while for the other experiments, the priors are artificially constructed. I feel more real-world experiments are needed, given that I'm doubtful as to whether the proposed algorithm is truly useful in practice as mentioned in the first point above.
- Why haven't you compared with the related work of Li et al. (2020)? It seems very similar to the proposed approach and is thus applicable in similar settings.

---

> ### Author Response · Authors · 2020-11-14
> **Response to AnonReviewer3**
>
> We are glad that the reviewer found our work intuitive and natural, the experiments comprehensive, and the paper well-written. We thank you for the constructive feedback and address the main concerns raised below.
>
> **Q1.** _Firstly_, asking the expert to specify a search space is intuitively easier for the expert, compared with asking them to give a distribution across the entire domain.
> _Moreover_, I think it's hard to justify why the proposed approach is better in practice than simply asking the expert for a reduced search space.
> **A1.** In PrBO, users define a search space just like in standard BO, while also allowing the injection of priors over the search space. Users could always choose a uniform prior at no extra work, but our experience is that users are often excited to define priors because they can exploit their knowledge about the black-box function and this requires only minimal extra work (a few minutes generally). Indeed some users are reluctant to use BO without being able to define simple priors like the one in PrBO. Concerning the second point, i.e narrowing the search interval further, would be a hard prior, making it difficult to recover from if it doesn't include the optimum. Practitioners rather have soft priors, 'hunches', which we support in PrBO.
>
> **Q2.** The proposed algorithm is a straightforward combination of PI, TPE, and EI. Therefore, the technical contribution may not be novel enough.
> **A2.** Although this may not be obvious at first glance, the implications of this seemingly obvious combination are major. PrBO makes three technical contributions that make it more flexible and efficient than existing state-of-the-art:
> 1. PrBO bridges the TPE methodology and standard BO probabilistic models, such as GPs, RFs or Bayesian NNs, instead of Tree Parzen Estimators only.
> 2. PrBO is flexible w.r.t. how the prior is defined, allowing previously hard-to-inject (e.g. exponential) priors.
> 3. PrBO gives more importance to the model as iterations progress, gradually forgetting the prior (Eq. 2) and ensuring robustness against misleading priors (Sec. 4.2).
>
> **Q3.** Figure 4: It seems that Spearmint consistently outperforms PrBO with weak prior. Does this mean that if you don't have a good prior, you should simply run standard BO?
> **A3.** In real-world applications it is usually not possible to know how strong the prior is, but as we show in practical applications, experts’ priors often tend to be a lot better than these weak priors. Nevertheless, the experiment in Figure 4 shows that PrBO still yields reasonable performance if the prior is indeed weak. In particular, it outperforms Spearmint in the SVM benchmark (Figure 9) and shows a head start in early iterations (~50 iterations) for the other benchmarks (which is a suitable property for applications with a small budget).
>
> **Q4.** There is only one real-world experiment (Section 4.4) where an expert is asked to specify a prior, while for the other experiments, the priors are artificially constructed.
> **A4.** Perhaps it is worth noting that our real-world experiments comprise three real-world applications, in all of which PrBO yielded state-of-the-art performance.
>
> **Q5.** Why haven't you compared with the related work of Li et al. (2020)? It seems very similar to the proposed approach and is thus applicable in similar settings.
> **A5.** PrBO presents fundamental differences that set it apart from the work of Li et al. (2020). Namely:
> 1. PrBO is agnostic to the model used, while the approach of Li et al. is restricted to GPs; this is important, since, for example, our real-world application has many categorical variables and RFs clearly outperform GPs for it.
> 2. PrBO provably recovers gracefully from misspecified priors, while the approach of Li et al. fundamentally trusts the prior and it never gets washed out by more data. For the case with a misspecified prior PrBO will thus clearly perform better.
>
> The code provided by Li et al. can not easily be extended to other benchmarks, which makes a direct comparison difficult, which is why we did not perform this empirical comparison. However, if the reviewer feels strongly that this comparison would be insightful, we can attempt to re-implement this over the next week and show a comparison against it.

---

### Official Review · AnonReviewer1 · 2020-10-28
**Official Blind Review #1**

**Rating:** 4
**Confidence:** 4

**Review:**



# summary

This paper provides an interesting setting and tries to incorporate prior
knowledge of inputs in BO. For example, in a specific application, an expert
knows the potential good region for some parameters in this problem. More
concretely, to incorporate expert knowledge in BO, the authors first specify a
"good" prior, which gives a large mass on more interesting regions. A "bad"
prior can also be specified, and a common choice is 1 - goodprior. These two
priors are built from expert knowledge.


# pros

1.  This paper provides an interesting direction in BO, i.e. incorporating
    prior knowledge. Moreover, this direction is very realistic, as in many
    applications, we do know some parameters are likely to take some specific
    values or fall into some regions.
2.  This paper also provides comprehensive experiments and verifies the
    advantages of incorporating prior knowledge in BO.


# cons

1.  My major concern is the construction of the "good" prior and the "bad"
    prior. If my understanding is correct, the "good" prior in all experiments
    are fitted using KDE. Based on the independence assumption, different
    parameters are fitted independently. The input data to KDE is {x_i}. To obtain a good fit, a large number of
    function evaluations is needed (many objective values are needed to get "good" x_i) and this violates the principles of BO in
    the first place, i.e. the objective function in BO is expensive and we
    cannot afford a large number of evaluations. In summary, to construct a
    reasonable "good" prior, we need a large number of function evaluations but
    we cannot afford it. If the "good" prior is not good, we then need a large
    number of functional evaluations to wash out such a prior in the following
    optimizing stage, violating the main principle of BO again.
2.  In Figure 2, the authors use a misspecified prior to show their proposed
    method is able to recover from a "wrong" prior. Indeed, this property is
    desirable. My question is if the prior might be wrong, why not use the
    conventional BO?
3.  Why not directly set the parameter bound to be the potentially interesting
    region in optimizing the acquisition? I understand directly setting the
    parameter bound in optimizing the acquisition function could rule out the
    optimal region if our prior knowledge is misspecified. If our prior
    knowledge is "correct", I guess this approach could have a good
    performance. Can the authors comment on this?
4.  Comparison with SMAC is lacking and I think this is a must. SMAC performs
    better than TPE in many problems.

Minor issues:
1. The claim: 10000x faster than random search, sounds like this is an ad instead of an academic paper.

Overall speaking, incorporating prior in BO is an interesting direction,
however, I am afraid this work do not have sufficient theoretical analysis and
there are some issues stated above on the prior in this work.

---

> ### Author Response · Authors · 2020-11-14
> **Response to AnonReviewer1**
>
> We are glad that the reviewer found our paper interesting and our experiments comprehensive. We thank you for the constructive feedback and address the main concerns raised below.
>
> **Q1.** Major concern on the construction of the "good" and the "bad" priors [confusion about the need of additional samples to construct the prior].
> **A1.** PrBO can use any prior that is available for an application. In practice, domain experts can directly specify the prior they have without requiring additional samples. This was the case for the priors in our real-world experiments in Section 4.4. The only reason why we used an expensive KDE prior in our synthetic experiments was to have a generic and fair prior that is not biased by our own expertise for the benchmarks. We emphasize that these KDE priors were only used to construct a way to properly *evaluate* our method; generating these priors is not part of our method, but our method takes these priors as a user-defined input. We do *not* propose using this approach to come up with a prior, but rather propose a method that allows using arbitrary priors that experts have.
>
>
> **Q2.** If the prior might be wrong, why not use the conventional BO?
> **A2.** It is not possible to know in advance whether a prior is wrong or right. Of course, we do not recommend to inject a wrong prior on purpose but to rather use conventional BO instead when we know the prior is wrong. However, domain experts often have a lot of knowledge about an application at hand and might believe that their prior is good even when it is not; in order to guard against such a case, in practice it is comforting to not have to worry about potentially having a wrong prior. Figure 2 shows that, in the event the expert mistakenly injects a misleading prior, PrBO is still able to forget the prior and find good configurations which is a desirable property of PrBO.
>
>
> **Q3.** Why not directly set the parameter bound to be the potentially interesting region in optimizing the acquisition?
> **A3.** It is not possible to define a prior that is certain to be correct. If the user is certain that a region will contain only poorly performing configurations, they can simply remove this from the search space in PrBO or conventional BO, but this approach would not be able to recover from misspecified priors. Instead, priors are used when users have a good intuition about promising configurations for a new application, but cannot be certain about them.
>
>
> **Q4.** Comparison with SMAC is lacking and I think this is a must. SMAC performs better than TPE in many problems.
> **A4.** We have added a comparison with SMAC to Figure 12 showing that PrBO is still the leading method.

---

### Official Review · AnonReviewer4 · 2020-10-29
**Strong idea with extensive evaluation and important applications**

**Rating:** 8
**Confidence:** 4

**Review:**

The paper presents a novel method to incorporate experts' knowledge into BO. This is done through introducing  Prior-guided Bayesian Optimization (PrBO). Different experiments where conducted to compare PrBO vs different baselines and to show the effect of the user provided priors in the cases where it is well-specified or mis-specified. The design of PrBO enables it to guide the search in the early iterations and as optimization progresses, more emphasis is given to the model and the effect of the prior is washed out.

Strong points:
- The paper presents a method for incorporating priors over good solutions in an elegant way.
- The method can benefit from the prior information and incase of misspecified priors, it can still recover.
- Extensive experiments were conducted.
- The paper is well written.
- This type of work is definitely needed to increase the adoption of BO methods.

Weak points:
- Although the experimental design is extensive and covers several aspects. I wish the experiments included one realistic experiment for tuning the hyperparameters of a famous architecture. This would have been a great example to demonstrate the benefits of this method versus the manual search used by most researchers. However, this is not a huge concern.

Clearly state your recommendation (accept or reject) with one or two key reasons for this choice.
- I recommend to accept this paper.
- Choice of priors affect the performance of BO. This paper provide the intuitive way to add priors by modelling probability of good configuration. This is what the user thinks. I am sure that this kind of work will encourage more researchers to use BO in their problems

Questions:
- It is clear how PrBO is incorporated in TPE, but what about GPs and RF? How is PrBO implemented in this case?
- Are the experiments in the main paper using GPs or TPE?
- Could you please explain more what 10,000×random search is?
- How are the weak and strong priors for PrBO generated?

---

> ### Author Response · Authors · 2020-11-14
> **Response to AnonReviewer4**
>
> We are glad that the reviewer found our work to be paramount to increase the adoption of BO. We thank you for the constructive feedback and address the questions raised below.
>
> **Q1.** It is clear how PrBO is incorporated in TPE, but what about GPs and RF? How is PrBO implemented in this case? Are the experiments in the main paper using GPs or TPE?
> **A1.** PrBO is agnostic to the model used, the approach presented in Section 3.2 works out-of-the-box for both RFs and GPs. All experiments in the main body and Appendix use GPs, except for the real-world experiments, where we use RFs for a fair comparison with state-of-the-art. (We stated this in Section 4.0 (GPs) and 4.4 (RFs), but this could admittedly be overread easily and we made it more explicit in 4.0.)
>
> **Q2.** Could you please explain more what 10,000x random search is?
> **A2.** 10,000x random search is a baseline that evaluates 10,000 uniform random samples for each sample that the other optimization methods evaluate. E.g. after 100 iterations, 10,000x random search evaluated 1,000,000 samples, while other methods evaluated 100.
>
> **Q3.** How are the weak and strong priors for PrBO generated?
> **A3.** The strong prior is a KDE fit on the best 10D out of 10,000,000D uniformly sampled points and the weak prior is a KDE fit on the best 10D out of 1,000D uniformly sampled points. We clarified this in Section 4.3. We emphasize that we only used such an artificial approach to generate the prior for the experiments to guarantee that our prior is not biased by our own expertise for the benchmarks we used, and that in a real application the priors are user-defined.

---

### Official Review · AnonReviewer5 · 2020-11-12
**Overclaiming and unsound probabilistic model**

**Rating:** 3
**Confidence:** 4

**Review:**

This paper presents a new method, PrBO, and it incorporates user knowledge on the location of potential optimum to the prior used for Bayesian Optimization. The authors claim that BO can be solved more efficiently using their method and showed promising empirical results on some low-dimensional benchmarks.

Disclaimer: I reviewed the same paper at Neurips. While I was happy to see that the authors have resolved some of term misusage issues and wordings, I felt somewhat disappointed that many good points raised by the reviewers there before were not addressed in the new version of the paper. IMHO, it still remains to be true that this paper needs a bigger surgery to be accepted, not only for Neurips or ICLR, but also other ML venues. And I hope the authors will consider the reviewers' opinions before submitting somewhere else.

Pros:
- Writing is mostly clear. Good clarity and good illustrations.
- I still think this is a very interesting idea and a quite promising direction to incorporate human knowledge in our used-to-be black-box machineries. This paper could set a good direction if it's done properly.

Cons:
- This wasn't the case of the previous version, but I found it unsettling that the 1st contribution point was "For the first time, user prior knowledge can be combined with standard BO probabilistic models, such as Gaussian Processes (GPs), Random Forests (RFs), and Bayesian Neural Networks." That is just bold and wrong. User prior knowledge is always considered in those models when selecting which kernel structure to use, how many trees would be sufficient, or if we need a convolution layer or not. Even in the narrowest definition of prior, this paper is a perfect demonstration of how to incorporate human knowledge in GPs: https://papers.nips.cc/paper/2015/file/4462bf0ddbe0d0da40e1e828ebebeb11-Paper.pdf. I strongly suggest not to ever claim one's the first, and when it's really needed, be very very specific. Otherwise there will be bad consequences. (See an extreme case related to Christopher Columbus.)
- It is still not clear what different probabilities mean to the authors. The authors described the prior P_g(x) as a prior on good points and P_b(x) as prior on bad points. What do those mean exactly? Could you define them correctly in the paper? And how does the GP-induced good/bad point probability interfere with the proposed ones?
- Footnote on Page 3: "P b(x) is not a probability distribution". What do you mean? In the text it says "prior distributionPg(x)".
- P_g is not independent for each x. Will that cause any issues to the method?
- If both the "prior" and "posterior" are both pseudo, what do they mean exactly? Or what are even approximating?
- Can the authors describe a coherent generative model of what they proposed?

I strongly suggest the authors carefully read the reviews from Neurips again.

---

> ### Author Response · Authors · 2020-11-14
> **Response to AnonReviewer5**
>
> We thank the reviewer for their review.
>
> The authors would like to clarify that, as reviewer 5 pointed out, another version of this paper was previously submitted to NeurIPS, where it received valuable feedback and was ultimately rejected. We appreciated the feedback received and, based on it, made major revisions of our paper to incorporate the points raised by the NeurIPS reviewers:
> - Added the TPE formulation to Section 2 (Background) to not conflate that with our model. We believe that this conflation was the main source of confusion in the NeurIPS version, and that the description of our model is now much clearer.
> - Added Proposition 1 to provide a formal motivation/understanding of the workings of our method and added a corresponding proof in Appendix C.2.
> - Reworded the text in several places to make it clearer that PrBO is a heuristic approach that is based on pseudo-priors and pseudo-posteriors.
> - Reworded our contributions to make it clearer what are the technical contributions of our work.
> - Added a comparison to state-of-the-art baselines with the same prior initialization in Appendix J showing that our approach still outperforms the state-of-the-art.
> - We made it clearer that PrBO does not assume or require independent univariate distributions and that we only assume univariate priors in our experiments since that’s what we expect the standard use-case to be.
>
> We sincerely hope that, in this discussion, we can focus on discussing *this* version of paper, and not any misunderstandings from the previous version. We believe that this version is far clearer, and in fact none of the 4 reviewers who only saw this new version seemed to be confused about the points that were confusing in the previous version.
>
>
> **Q1.** The authors showed promising empirical results on some low-dimensional benchmarks.
> **A1.** Thanks. We would like to point out that the MD grid benchmark is 10-dimensional and the XGBoost benchmark is 8-dimensional, which is actually quite high-dimensional for the BO literature.
>
> **Q2.** [Unsettled about 1st contribution point]
> **A2.** We agree with the reviewer that this claim was not properly worded. We have now clarified our intention in the text, i.e., that our work combines user priors in the form of flexible probability distributions over which inputs are expected to perform well with any of the standard BO models.
>
> **Q3.** It is still not clear what different probabilities mean to the authors. The authors described the prior P_g(x) as a prior on good points and P_b(x) as prior on bad points. What do those mean exactly? Could you define them correctly in the paper?
> **A3.** We would like to ask for a more concrete proposal on how to improve the paper wrt these questions. For us and for the other reviewers, it is clear that P_g(x) defines a probability of x being a good configuration. Furthermore, the math in the main paper and in the appendix fully define how our proposed algorithm works and in contrast to our NeurIPS submission, we now state the heuristic parts of our approach more explicitly.
>
> **Q4.** How does the GP-induced good/bad point probability interfere with the proposed ones?
> **A4.** We note that the GP prior and P_g(x) provide orthogonal ways to input prior knowledge: the GP prior is typically a *zero-mean* GP and therefore only specifies our expectations about the smoothness of the function and does *not* interfere with P_g(x), which only specifies knowledge about regions with good configurations. The GP posterior of course is not zero-mean, and we use it to define M_g, which is combined with P_g.
>
> **Q5.** Footnote on Page 3: "P_b(x) is not a probability distribution". What do you mean? In the text it says "prior distribution P_g(x)".
> **A5.** We refer to the full Footnote 1 on Page 3 that states: “We note that for continuous spaces, this P_b(x) is not a probability distribution, and therefore only a pseudo-prior, as it does not integrate to 1. For discrete spaces, we normalize P_b(x) so that it sums to 1 and therefore is a proper probability distribution and prior.” We followed the proposal of the NeurIPS reviewers to call this a pseudo-prior.
>
> **Q6.** P_g is not independent for each x. Will that cause any issues to the method?
> **A6.** While this was unclear in the NeurIPS version, this is one of the points we made very clear for this version, please see Section 3.1. PrBO very explicitly does *not* require P_g to be independent. Users *can* define multivariate distributions for P_g. Since in practice, we expect that most users will want to specify univariate priors instead for convenience (just like search spaces are usually defined as a box and not, e.g., as an ellipse), we focus the experiments in our main paper on the univariate case. We show in Appendix G that PrBO also works with multivariate priors.

---

> > ### Author Response · Authors · 2020-11-14
> > **Response to AnonReviewer5 (cont.)**
> >
> > **Q7.** If both the "prior" and "posterior" are both pseudo, what do they mean exactly? Or what are even approximating?
> > **A7.** We used this nomenclature as proposed by the NeurIPS reviewers. They are score distributions that are inspired by the Bayesian way of combining a prior with a likelihood to obtain an (unnormalized) posterior.
> >
> > **Q8.** Can the authors describe a coherent generative model of what they proposed?
> > **A8.** There are two probabilistic models:
> > - The standard probabilistic model of BO, with a prior over functions p(f), updated by data (x\_i,y\_i)\_{i=1}^t to yield a posterior over  functions p(f|(x\_i,y\_i)\_{i=1}^t), allowing us to quantify the probability M\_g(x)  = p(f(x) < f\_\gamma | x, (x\_i,y\_i)\_{i=1}^t) = \Phi( … ) from Equation 2
> > - The TPE-like generative model that combines p(y) and P(x|y) instead of directly modeling p(y|x);
> >
> > With regards to the generative model, we simply use TPE’s generative model. The only thing we change is to replace its KDE with an M\_g computed from BO’s model. This is done using Equation (2), which bridges these two models by using the probability of improvement from BO’s standard model as the probability M\_g(x) in TPE’s model. Ultimately, this is a heuristic since there is no formal connection between the two probabilistic models. However, we believe that the use of BO’s familiar, theoretically sound framework of probabilistic modeling of p(y|x), followed by the computation of the familiar PI formula is a very intuitive choice for obtaining the probability of an input achieving at least a given performance threshold -- exactly the term we need for TPE’s M\_g(x).
> > We have added this clarification to Section 3.2.
> >
> > We thank the reviewer again for their work and are looking forward to a constructive discussion of our current paper.

---

### Decision · Program_Chairs · 2021-01-07
**Final Decision**

**Decision:**

Reject

**Comment:**

This paper was quite contentious.  While there is clearly promise in the method and the idea, and reviewers appreciate the importance of encoding non-trivial prior knowledge into BO, three reviewers express major concerns regarding the presentation (including worries about over-claiming contributions), the specification of the probabilistic model as well as to some extent about the experimental evaluation.  Lastly, the title appears to be a kind of pleonasm - arguably, the key point of using a Bayesian prior is to be able to encode prior knowledge.  The authors consider a different form of prior than perhaps usually meant in BO (in a generative rather than discriminative sense), but the terminology is still confusing.